# Memorisation in Machine Learning: A Survey of Results

**Dmitrii Usynin**                                                                                      *du216@ic.ac.uk*
*Department of Computing, Imperial College London*
*Institute for AI in Medicine, Technical University of Munich*

**Moritz Knolle**                                                                                       *m.knolle@tum.de*
*Institute for AI in Medicine, Technical University of Munich*
*Konrad Zuse School for Excellence in Reliable AI*

**Georgios Kaissis**                                                                                    *g.kaissis@tum.de*
*Institute for AI in Medicine, Technical University of Munich,*
*Institute for Machine Learning in Biomedical Imaging, Helmholtz Munich*

**Reviewed on OpenReview:** *hhttps://openreview.net/forum?id=HVWODwbrFK*

## Abstract

Quantifying the impact of individual data samples on machine learning models is an open research problem. This is particularly relevant when complex and high-dimensional relationships have to be learned from a limited sample of the data generating distribution, such as in deep learning. It was previously shown that, in these cases, models rely not only on extracting patterns which are helpful for generalisation, but also seem to be required to incorporate some of the training data more or less *as is*, in a process often termed *memorisation*. This raises the question: if some memorisation is a requirement for effective learning, what are its privacy implications? In this work we consider a broad range of previous definitions and perspectives on memorisation in ML, discuss their interplay with model generalisation and their implications of these phenomena on data privacy. We then propose a framework to reason over what memorisation means in the context of ML training under the prism of individual sample's influence on the model. Moreover, we systematise methods allowing practitioners to detect the occurrence of memorisation or quantify it and contextualise our findings in a broad range of ML learning settings. Finally, we discuss memorisation in the context of privacy attacks, differential privacy and adversarial actors.

## 1 Introduction

Machine learning (ML) models require access to large amounts of high-quality, diverse and well-curated data to perform well in a variety of tasks ranging from biomedical imaging (Seo et al., 2020; Rueckert & Schnabel, 2019) to text generation using large language models (LLMs) (Nijkamp et al., 2022; Taylor et al., 2022; Alayrac et al., 2022). Such data is often sensitive in nature, mandating that its disclosure be avoided. However, with the advent of generative ML, it has become apparent that models often reproduce samples from their training datasets almost verbatim (Ippolito et al.; Carlini et al., 2019b; 2021), thus posing potential privacy risks for data owners. This phenomenon is not new to the ML privacy community, as it has been known that models trained without the use of privacy-enhancing techniques such as differential privacy (DP) are prone to inferences about their training data, such as membership inference (MIA) (Shokri et al., 2017) or data reconstruction attacks (Zhu et al., 2019). However, these observations in generative models have led to a recent spike in research trying to uncover (1) whether and to what extent models actually *contain* their training data (to then regurgitate it) and (2) which training samples are more prone to this.

The aforementioned phenomenon is (often informally) termed *training data memorisation*. Memorisation is often viewed (and taught) as being the opposite pole of generalisation (that is, committing data samples to

storage rather than learning general patterns from the training data). Obviously, this distinction is somewhat artificial, as it is all but impossible to delineate where general patterns end and the storage of actual data samples begins. Beyond this fact, it has empirically been shown that –while models are able to learn data representations which are useful to the task at hand– they are also entirely capable of fitting random input-output associations such as purely random labels in the context of supervised learning (Zhang et al., 2017). In addition, while supervised ML models are usually able to perfectly fit to their training data, this often does not translate to a commensurate generalisation performance on unseen data.

Additionally, certain types of ML models, such as support vector machines or $k$-nearest neighbours, exhibit a learning process which is in its entirety based on storing data samples followed by a subsequent look-up process. It thus becomes evident that the delineation between memorisation and generalisation may –in fact– be moot. To the contrary, it appears that *both*, memorisation and generalisation, are crucial to the learning process of ML models. Prior works lend credence to this fact. An early work by Chatterjee (2018) demonstrates that a network of support-limited lookup tables can memorise enough patterns to be able to generalise to those it has never previously seen. Additionally, Brown et al. (2021) find that specific tasks can only be successfully learned through memorisation of portions of the training data. Moreover, the seminal works of Feldman & Zhang (2020); Feldman (2020); Zhang et al. (2021a) demonstrate that memorisation and generalisation not only co-occur, but that the former is actually a prerequisite for the latter if one aims to train a model of (close-to) optimal utility. Several prior works also equate memorisation to overfitting, the phenomenon where, during training, the model's performance on unseen data starts deteriorating after initially having increased, pointing to an over-accommodation of the model to its training data.

These introductory remarks reveal that there is not only a lack of terminological clarity when discussing memorisation and generalisation in ML, but also that there exist a multitude of –partially conflicting– definitions. We thus identify a requirement for a systematisation of prior works describing these phenomena in various sub-domains of ML. Moreover, we contend that such a systematisation must also tackle specific related points of importance:

- Pinpointing which samples are (more) prone to memorisation;

- Quantifying, rather than merely detecting the presence of memorisation;

- Identifying the implications of memorisation in terms of data privacy;

- Discussing techniques aimed at preventing or diminishing memorisation and their consequences on the model's performance.

Such a work will not only aid in the disambiguation of the multiple existing definitions of memorisation, but also assist practitioners in choosing the techniques to identify, measure and possibly reduce memorisation in their operational scenario or use-case. We provide a summary of all techniques discussed in this work in Table 1.

In this paper, we attempt the aforementioned systematisation. Our work is structured as follows:

- We outline a formalised definition of memorisation in machine learning and discuss how it can be quantified directly (Sections 2, 3 and 4.1);

- In addition, we discuss notions which are ostensibly related to (but otherwise disjoint from) memorisation (Section 5);

- We then discuss how, when and in which part of the learning protocol does memorisation get induced (Sections 6 and 7.1);

- We discuss the implications of memorisation on the privacy of the individuals whose data is used to train the model (Sections 8.1 and 9).

- Finally, we provide guidelines for ML practitioners and outline promising future work directions (Sections 10 and 11).

As ML encapsulates a large variety of learning tasks, modalities and domains, we discuss the implications these may have under the prism of memorisation. Specifically, we discuss the implication of memorisation in different modalities (imaging and textual) in Sections 8.1 and 8.2; learning settings (discriminative vs generative) in Section 8.2 and data regimes (centralised and distributed) in Appendix A. We additionally provide some practical guidelines on context-specific memorisation for the practitioners to have a better understanding of what memorisation can entail when working with sensitive data in Section 10.

| Techniques | Section | Challenges | Measures memorisation? | Examples |
|---|---|---|---|---|
| Re-training methods | 4.1 | - Poor scaling with the dataset size | Yes | Feldman & Zhang (2020); Zhang et al. (2021a) |
| Influence functions | 4.1 | - Poor scaling with the model size
- Unsuitable for deep learning | Partially [1] | Koh & Liang (2017); Guo et al. (2020); Grosse et al. (2023); Kounavis et al. (2023) |
| Data valuation techniques | 4.2 | - Poor scaling with the dataset size
- Often non-transferable across settings
- Measure value of data, not memorisation | Partially [2] | Ghorbani & Zou (2019); Hammoudeh & Lowd (2022a) |
| Gradient-based methods | 5.1 | - Model- and data-dependent
- Poor scaling with the model size
- Capture various phenomena at once | No | Chen et al. (2020b); Zhu et al. (2022); Li et al. (2021); Garg & Roy (2023a); Mueller et al. (2022); Agarwal et al. (2022) |
| Information-theoretic approaches | 5.2 | - Unintuitive to the user
- Poor scaling with the model size
- Concern information flow, not memorisation | No | Shwartz-Ziv (2022); Goldfeld et al. (2018); Goldfeld & Greenewald (2021); Xu et al. (2020) |
| Sample difficulty estimation | 5.3 | - Model-, data- and task-dependent
- Describe several related phenomena at once
- Often non-transferable across settings | No | Chen et al. (2021); Baldock et al. (2021); Garg & Roy (2023b); Carlini et al. (2019a) |
| Induced memorisation measurement | 7.1, 9.2 | - Can affect the final model
- Model-, data- and task-dependent
- Can impose unrealistic assumptions
- Often measure leakage, not memorisation | Partially [3] | Carlini et al. (2019b); Tirumala et al. (2022); Thakkar et al. (2020); Hartley & Tsaftaris (2022); Carlini et al. (2021) |

Table 1: Summary of techniques used to identify data points which are more likely to be memorised. (1) - While influence functions *can* be used to calculate self-influence, these rely on assumptions which are often not satisfied in deep learning (e.g. strong convexity); (2) - Shapley values were previously described in Zhang et al. (2021a) as similar to calculation of self-influence; (3) - These can be used to measure the worst-case unintended memorisation, which frequently encapsulates memorisation, extractability and the ability of the model to leak data. Distinguishing between these three notions is a non-trivial task.

## 2   Preliminaries

Most ML models can be viewed as parameterised functions $f_\theta(\cdot)$, which, given some input $x$ and parameter values $\theta$, produce a corresponding output $f_\theta(x)$. Using a training dataset $S \sim \mathcal{D}$, where $\mathcal{D}$ is the data-generating distribution, many ML algorithms can be reduced to finding an optimal parameter setting $\theta^*$ (and the corresponding function $f_{\theta^*}$) that minimises a loss function $L$ over $S$:

$$\theta^* := \arg\min_\theta \sum_{i=1}^n L(f_\theta(x_i), y_i), \tag{1}$$

which is often called empirical risk minimisation (ERM). Based on the form that $S$ takes, we now broadly categorize ML methods into: (I) *supervised learning*, where the training dataset $S$ contains both inputs $x$ as well as ground-truth outputs/labels $y$ and (II) *generative learning/modelling*, where no corresponding example outputs are available, but we wish to learn the general structure of $x$. We use the terms *supervised* and *discriminative* interchangeably in this work.

For the rest of this paper, will omit $\theta$ (unless explicitly necessary) and denote the result of this procedure (commonly referred to as model training/fitting) as applying training algorithm $A$ to the dataset $S$ as $f \leftarrow A(S)$.

Finally, we frame the goal of ML as obtaining a model $f$ that generalises to unseen data. This is measured by the generalisation performance/error, which for supervised learning is defined as follows:

$$\text{err}_{\text{gen}}(f) := \mathbb{E}_{(x,y)\sim\mathcal{D}}[L(f(x), y)]. \tag{2}$$

For the generative learning setting, $L$ takes only one input, as $S$ contains no labels $y$. Note that formally, generalisation performance must be measured over the entirety of the data-generating distribution $\mathcal{D}$. In practice, it can only be estimated as an empirical average over a (finite) test set. Accurate estimation of $\text{err}_{\text{gen}}$

thus hinges on the choice of a representative test set that is distinct from the training dataset. Throughout, we will use the terms *sample*, *instance* and *data point* synonymously.

## 3 A formal definition of memorisation

For a long time, memorisation lacked a precise definition and the term was commonly used loosely to refer to a variety of phenomena. In this section, we discuss the definition of Feldman (2020), who presented the first unified formulation and theory of memorisation in ML. We note that in this work we primarily concentrate on the underlying intuition behind the definition of Feldman (2020), rather than its specific operationalisation. While various other definitions have previously been proposed (and we discuss them in great detail in the following sections), we identify that the one proposed in Feldman (2020), which is based on the notion of influence as the only one which quantifies the phenomenon of memorisation itself, rather than measuring some related phenomena often associated with samples which can be memorised. We additionally note that due to the choice of the influence-based formulation, this definition is also a) modality- b) training setting- and c) model-agnostic, making it an attractive generic formulation of memorisation. This is primarily because the notion of *influence* can be easily extended to any learning scenario allowing the user to select which metric is used to quantify how the presence or absence of a training point affects the resulting utility (e.g. per-sample accuracy in classification or log-perplexity in language modelling). In contrast, most of the previously proposed definitions which we discuss in this manuscript are either data- or model-specific (e.g. quantifying memorisation via canaries); only apply to specific training settings (e.g. fitting of random labels) or capture a different, albeit related phenomenon (e.g. memorisation through overfitting). Fundamentally, this definition is also very intuitive: if we (i.e. humans) memorise a sample, we are expected to be able to make a more accurate prediction on it compared to a setting where we have never seen this exact sample before and only extrapolate from our knowledge of similar-looking data. Methods which do not directly measure memorisation (but, instead, certain factors leading to it) all seem to leverage this fact: the samples which fall under this behaviour are distinct and individual, (or informally "special"). This unifies the previous work on these metrics (which we later describe as methods that ostensibly measure memorisation) under this common thread. This makes Feldman (2020) formulation of memorisation a very attractive definition to describe the **phenomenon** of memorisation (with an intuitive method to quantify it regardless of the learning setting).

In the framework of Feldman (developed further in Feldman & Zhang (2020); Zhang et al. (2021a)), memorisation is framed as the impact a particular sample has on its own prediction (known as the *self-influence*). Formally, it is defined as the difference in (expected) performance on the sample (with index) $i$ when sample $i$ is included in the training dataset $S$:

$$\text{mem}(A, S, i) := \underbrace{\mathbb{E}_{f \leftarrow A(S)} \left[ \text{M} \left( f(x_i), y_i \right) \right]}_{\textbf{performance on } i \textbf{ when } i \in S} - \underbrace{\mathbb{E}_{f \leftarrow A(S \setminus i)} \left[ \text{M} \left( f(x_i), y_i \right) \right]}_{\text{performance on } i \text{ when } i \notin S} . \tag{3}$$

Here, the expectation is taken over the randomness of the training algorithm $A$. M refers to some suitable performance metric, (e.g. accuracy) and $S \setminus i$ is the training dataset with sample $i$ removed. Note that Eq. (3) is context agnostic and can be applied to a generative settings by picking a performance metric $M$ that only takes a single input argument (i.e. without including the ground truth labels). The algorithm here represents the training process of the model, during which memorisation occurs, but it is ultimately the model itself that memorises. Note that a naive calculation of Eq. (3) is computationally expensive. Efficient sub-sampling estimators were proposed and are discussed in Section 4.

> **Key Point**
>
> *Memorisation* is formally defined as the influence a sample has on its own (correct) prediction (self-influence).

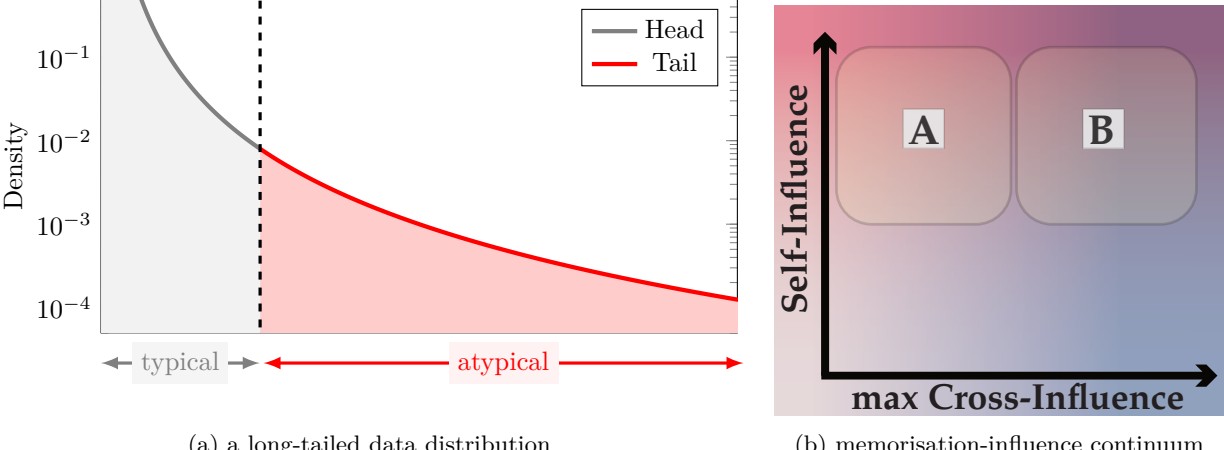

(a) a long-tailed data distribution        (b) memorisation-influence continuum

Figure 1: Long-tailed distributions have probability density that decays very slowly for extreme (atypical) values (a). It can thus produce samples that are so distinct from typical samples that they are required to be memorised to allow for generalisation to other similar samples. Sub-figure (b) shows a schematic visualisation of the memorisation-influence continuum. While all samples from the tail of the data distribution are likely memorised during training (large self-influence), useless, low-quality or mislabeled samples (A) are indistinguishable from useful samples that significantly influence the correct prediction of at least one test sample (large max cross-influence) (B).

### 3.1 Influence and the long-tail theory

In the same work Feldman (2020) proved that memorisation is a required component of learning. This *long-tail theory*, contends that close-to-optimal generalisation performance on long-tailed data distributions (see Fig. 1a) necessitates the memorisation of rare and "atypical" samples from the tail of the distribution. This is due to the fact that, in long-tailed distributions, samples from low-density regions (the tail) differ extremely from samples from high-density regions (the head). Thus, memorisation of these atypical samples is necessary to generalise optimally to other atypical samples at test time.

The long-tail theory (developed further in Feldman & Zhang (2020); Brown et al. (2021)), was supported by prior observations that naturally occurring data distributions commonly have long tails (Babbar & Schölkopf, 2019; Zhu et al., 2014; Van Horn & Perona, 2017; Yang et al., 2022). Further empirical evidence in direct support of this theory was presented in Feldman & Zhang (2020); Zhang et al. (2021a), which demonstrated that (for commonly used large-scale vision and text datasets), a substantial proportion of training examples have an out-sized impact on the model's performance on few specific test examples. The (sample-level) measure used to demonstrate this phenomenon was termed *cross-influence*, that is, the impact a training sample $x_i$ has on the prediction of a test sample $x'_j$:

$$\text{infl}(A, S, i, j) := \underbrace{\mathbb{E}_{f \leftarrow A(S)}[\text{M}(f(x'_j), y'_j)]}_{\textbf{performance on } j \textbf{ when } i \in S} - \underbrace{\mathbb{E}_{f \leftarrow A(S \setminus i)}[\text{M}(f(x'_j), y'_j)]}_{\textbf{performance on } j \textbf{ when } i \notin S} , \tag{4}$$

whereby self-influence (mem) is recovered when $i = j$ and $M$ is some suitable performance metric that returns higher values for better predictions. Thus, a positive infl value indicates that training sample $i$ improves the prediction on test sample $j$ when $i \in S$. If $M$ is taken to be a risk/loss function, equivalent behaviour can be achieved by simply negating all infl values.

Fig. 1b visualises the memorisation-influence continuum. The continuum illustrates that while all samples from the tail of the distribution are likely to be memorised (and thus exhibit high self-influence), not all

of them have a significant impact on at least one test sample (A vs. B). Further, a direct consequence of the long-tailed nature of data distributions is the fact that, at training time, low-quality or mislabelled samples (A) are statistically indistinguishable from useful representative samples of rare sub-populations (B) Feldman (2020).

> **Key Point**
>
> When a data distribution is *long-tailed*, memorising samples from the tail of the distribution can help a model generalise.

### 3.2 Prior efforts to capture memorisation

There have previously been many attempts to quantify memorisation in ML. However, we stress that while many of these phenomena have previously been labelled as "memorisation" or claim to identify samples which are "memorised" by the model, **none** of these can be used as a formal and modality-agnostic definition of memorisation. In this section we outline some of the most well-known phenomena, which, while related to memorisation are **not** suited to quantify this notion. We outline additional methods which have previously been used to quantify memorisation in Table 1.

**Memorisation as overfitting** A number of prior works (Olatunji et al., 2021; Chen et al., 2020a; Kuppa et al., 2021; Leino & Fredrikson, 2020; Veale et al., 2018; Hilprecht et al., 2019; Mehta et al., 2020) draw direct connections between memorisation and overfitting, defining memorisation of a given model as its generalisation gap (i.e. the empirical measure of overfitting). Formally, the generalisation gap of a model $f$ is defined as:

$$\text{err}_{\text{gap}}(f) := \underbrace{\text{err}_{\text{gen}}(f)}_{\text{generalisation error}} - \underbrace{\sum_{i=1}^{n} L(f(x_i), y_i)}_{\text{empirical (training) error on } S} \tag{5}$$

This quantity is task-agnostic and it can be estimated using a representative test set, making it a very attractive tool for empirical memorisation measurement. However, in the light of evidence from Feldman & Zhang (2020); Zhang et al. (2021a), showing that the memorisation of specific training examples actually increases generalisation performance, this definition now appears dated. In addition, the measure above does not allow for the identification of individual memorised samples, being computed as an average. Finally, recent works demonstrate that memorisation usually precedes overfitting (Carlini et al., 2019b; Tirumala et al., 2022).

> **Key Point**
>
> Memorisation **cannot** be reduced to overfitting. Overfitting indicates a lack of generalisation, whereas memorisation was shown to improve generalisation.

**Fitting random labels** Zhang et al. (2017) demonstrated that even relatively small neural networks (by today's standards) can perfectly fit large datasets with randomised labels or even completely random data. As the correct prediction of a random label is impossible without memorisation, randomised labels are commonly used to study qualitative aspects of memorisation behaviour in ML models, such as whether it can be localised to specific regions in the model or when it temporally occurs during training. These topics are discussed in more detail below in Section 6.

However, as this formulation relies on the presence of labels, it can often be challenging to quantify memorisation in settings where these are absent (e.g. generative modelling). We, thus, turn to influence functions, which allow for such quantification to be performed efficiently in the following section.

> **Key Point**
>
> Many deep neural networks have sufficient memorisation capacity to fit large datasets of completely random input-output associations.

## 4 Measuring memorisation through influence estimation

Influence analysis dates back to (Cook, 1977) and aims to offer a data-centric approach to explaining a model's predictions by apportioning credit (positive or negative) to individual training data samples. If applied to quantify the impact that a single training data point has on the prediction of a single test sample, it is often called *point-wise influence* and is equivalent to the quantity (cross-influence) introduced in Eq. (4). This canonical definition of influence (commonly known as leave-one-out (LOO) influence) is computationally expensive. Its simple Monte Carlo estimation requires $\Omega(n)$ runs of training algorithm $A$ to convergence. Fortunately, substantial research effort has gone into developing efficient methods for influence estimation, which will be discussed next.

As covered previously in Section 3, memorisation can be quantified through *self-influence*. Thus, all methods discussed in this section can be used as valid approaches to estimate memorisation (if applied correctly to measure a sample's influence on its own prediction). We omit a comprehensive discussion of this topic and refer to Hammoudeh & Lowd (2022b) for an extensive survey on influence and its estimation.

### 4.1 Efficient influence estimation

Estimation of influence can be broadly categorised into re-training-based methods and influence functions.

**Re-training-based methods** approximate Eq. (4) directly through simple Monte Carlo sampling. To circumvent the aforementioned computational complexity, Feldman & Zhang (2020) proposed a *sub-sampling estimator*. This method simultaneously leaves out multiple training samples randomly (instead of just one) and records each sample's membership and the result, which are subsequently averaged to obtain a smoothed estimate of a sample's influence. Correspondingly, the number of required training runs is drastically reduced, as every run of Algorithm $A$ now contributes to the influence computation of many samples, rather than just one. Authors showed that this estimator is statistically consistent and that its error is bounded (with high probability) by the sub-sampling rate and the number of repetitions per randomly sampled subset (Feldman & Zhang, 2020). In practice, this method allows for accurate influence estimation (self/cross-influence) in multiple hundreds or a few thousand runs on large datasets (Feldman & Zhang, 2020; Zhang et al., 2021a). A (somewhat) similar approach was proposed by Jiang et al. (2020), termed the *per-sample generalisation*, measured using a consistency score, which estimates the expected accuracy of the model on a single sample, by comparing the performances of models trained on sets of varying sizes sampled from the data distribution. This approach, however, also suffers from computational complexity and a number of proxy metrics, such as the pairwise sample distance were proposed to approximate this value.

**Influence functions** were first ported from the domain of robust statistics to deep learning through the seminal work by Koh & Liang (2017). Using a classical result from robust statistics (Cook & Weisberg, 1982), Koh and Liang showed that a change in loss (due to removal of a sample) can be approximated by the inverse empirical Hessian (matrix of second-order derivatives w.r.t model parameters). This means that no model re-training is required. Intuitively, through a second-order Taylor series expansion, the effect of a small perturbation (assumed to be induced by removing a single sample) on the loss function can be approximated. Hessian matrices are infeasible to compute for large models, requiring memory which is square in the number of network parameters. However, efficient approximation techniques have been proposed to tackle this issue (Guo et al., 2020; Schioppa et al., 2022), allowing for the application of influence functions even to large language models (Grosse et al., 2023; Kounavis et al., 2023).

Re-training-based approaches, given enough computational resources, are able to provide reasonably accurate influence estimates, even for large datasets such as ImageNet (Feldman & Zhang, 2020). On the other hand, influence functions have received substantial criticism (Bae et al., 2022; Basu et al., 2020; Schioppa et al.,

2023), as key assumptions of the underlying theory, e.g. strong convexity and positive definiteness of the Hessian (amongst others), are not satisfied in the context of deep neural networks. Furthermore, multiple works have shown that influence function values correlate poorly with LOO influence values (Bae et al., 2022; Basu et al., 2020), an effect that seems to be aggravated with network size and depth. Thus, based on this evidence, we deem re-training-based methods the preferred choice for estimating memorisation if the available computational resources allow for it.

An alternative approach to the two above-mentioned methods relies on using the so-called "representer theorem", allowing the model owner to decompose the prediction into individual contributions (i.e. influences) from each data point (Yeh et al., 2018). This method has however been referred to as "overly reductive" (Yeh et al., 2022) in a follow-up work by the same authors as it's limited to detecting changes in the final layer only.

> **Key Point**
>
> Memorisation can be measured efficiently through influence estimation techniques, but care is required to make sure such estimates are accurate.

## 4.2 Connections to data valuation

Data valuation is concerned with assigning value to data points to determine equitable monetary compensation for data owners. It is closely connected to the concept of influence i.e. when influence is applied to quantify the impact of a single training sample on many test samples. In fact, the notion of the aforementioned LOO influence is also frequently used in data valuation. Note also that, any data valuation measure can be used as a point-wise influence measure (Hammoudeh & Lowd, 2022a) which means that conversely any data valuation measure can be used to measure self-influence. As influence measures were not originally designed to be equitable, substantial effort has gone into the game-theoretic analysis and improvement of data valuation/influence measures which satisfy a number of axioms aimed at improving data valuation equitability.

Arguably, the most popular valuation metric is the Shapley value (Ghorbani & Zou, 2019) which quantifies the expected marginal contribution of individual data points while considering all possible subsets of the dataset and their interactions. In many cases, the data samples which are "valued higher" are often the same ones that are more likely to be memorised (Zhang et al., 2021b). While such improved (re-training-based) data valuation/influence estimation methods have a clear advantage for equitable data owner compensation, the relevance of such game-theoretically motivated axioms for the quantification of memorisation remains unclear. Concretely, it has been argued by Zhang et al. (2021a), that LOO influence (which does not model data point interactions), is to be regarded as the preferred metric for measuring memorisation since it allows for making *causal* statements about the memorisation of a data point. Further, the authors argue that it's the insensitivity of LOO influence to data duplicates (a behaviour previously criticised in the context of data valuation (Ghorbani & Zou, 2019)) which renders it a suitable metric for measuring the memorisation of rare data points. As a result, while there are some clear overlaps between the domains of memorisation and data valuation, there are still many open questions (particularly on the relationship of the valuation metrics, which often concentrate on the cross-influence and the self-influence discussed in the context of memorisation) making the use of data valuation techniques for quantification of memorisation problematic.

> **Key Point**
>
> Game-theoretic data valuation metrics assign higher values to samples which are often more likely to be memorised, but are not a direct proxy for memorisation.

# 5 Related metrics incorrectly described as memorisation

Beyond the techniques which can be used to directly estimate memorisation introduced in Section 3 (i.e. through self-influence), there exists a large number of methods which attempt to rank training samples based on their "importance" to the model. Here the term importance encapsulates concepts such as difficulty of fitting, error/gradient magnitude, etc.. While, ostensibly, some of these metrics are associated with memorisation (and are often described as its computationally inexpensive approximations), we note that none of them estimate self-influence directly. We, therefore, postulate that **none** of the metrics discussed below can **directly** quantify memorisation. We note, however, that some of the metrics from these various domains often correlate with how much memorisation individual data points can experience, these cannot be used as direct "proxies" of this phenomenon.

## 5.1 Gradient-based influence proxies

The methods introduced in this section leverage the gradient of the loss function evaluated at individual samples, either with respect to the model weights or to the inputs themselves. Recall that the gradient is a linear/first-order approximation to the effect of a sample on the loss. The main benefit of the techniques discussed below is that they are easy to implement and computationally efficient.

**Gradients w.r.t. model parameters** Arguably, the most widely used gradient-related metric is its magnitude, i.e. the $\ell_2$-norm with respect to the model's parameters (Chen et al., 2020b; Amiri et al., 2021; Lai et al., 2021). Some works claim that the gradient norms of individual data points are proxies for memorisation (as, similarly to loss values, they tend to decrease over the course of training, highlighting that samples get "gradually" memorised) (Zhu et al., 2022; Katharopoulos & Fleuret, 2018; Li et al., 2021; Xue et al., 2021). However, while higher gradient norms (similarly to loss values) are usually associated with difficulty when making predictions on individual data points (often correlated with samples of higher influence), this is not a direct representation of memorisation (Zhu et al., 2022; Li et al., 2021). Moreover, even when viewing these metrics through the lens of adversarial susceptibility, there is no clear causal relationship between the amount of information memorised about individual samples; information contained in these samples; and the information exposed by the model. The gradient norm is also central to differentially private stochastic gradient descent (Abadi et al., 2016) and especially in approaches employing individual privacy accounting (Feldman & Zrnic, 2021; Yu et al., 2023; Koskela et al., 2022), where the gradient norm is proportional to the individual's privacy loss. Despite that fact, and while larger gradient norms *can* be associated with higher attack susceptibility, this is not always observed, implying that other factors may be involved (Usynin et al., 2023a; Geiping et al., 2020; Balle et al., 2022).

**Gradients w.r.t. inputs** Orthogonal to the aforementioned methods are techniques which utilise gradients with respect to the inputs. The most notable technique in this category is the *variance of gradients* by Agarwal et al. (2022), which combines tracing the gradient values with respect to the inputs during training with the computation of their variance over the time axis. A higher variance is claimed to imply that the sample is more atypical and is, hence, more difficult to learn. Thus, the authors state that samples with high variance of gradients are also more prone to memorisation. A distinct line of work (Mueller et al., 2022; Hannun et al., 2021) from the domain of privacy-preserving ML also used gradients (or second-order derivatives) with respect to inputs to measure privacy loss. These can be used to identify samples which have "more revealing" input features and be more susceptible to attacks on privacy, allowing the data owner to identify the privacy risks associated with individual training records.

We note that, techniques based on the second-order derivatives, which estimate the curvature of the loss function with respect to individual inputs (instead of weights) have also been used to measure memorisation (Garg & Roy, 2023a). Moreover, more computationally efficient techniques not requiring second-order derivatives have been developed, such as techniques which *trace* the evolution of the loss function's value or the (first order) gradient through training (Pruthi et al., 2020; Hammoudeh & Lowd, 2022a). These metrics can also be used to either A) identify samples which are more challenging for the model to learn from or B) be used to approximate the aforementioned influence functions (which are, in turn, often used to quantify memorisation). While these are conceptually similar to the efficient influence estimations proposed in Koh & Liang (2017), the subtle difference between the two concerns which type of curvature the authors discuss.

For influence estimation, the derivatives are taken with respect to the model weights, compared to methods such as (Garg & Roy, 2023a) (or a first-order approximation proposed in Agarwal et al. (2022)), where the derivatives are taken with respect to the model inputs).

Overall, many properties of training data which can be efficiently represented through the gradients (e.g. the privacy loss) are often correlated to the magnitude of memorisation of that training sample. Therefore many of the samples that possess e.g. higher gradient norms are often those that are also more likely to be memorised. However, these links are mostly correlational as to-date there has not been any empirical or theoretical evidence to show that there is a causal relationship between the characteristics of the gradient and the memorisation of individual samples. Hence, we argue that the gradient-derived metrics **cannot** be used as direct approximations of memorisation. Several works point to a connection between neural network geometry and memorisation, which we thus regard as a promising area for future research (Stephenson et al., 2021; Garg & Roy, 2023a; Ravikumar et al., 2024).

> **Key Point**
>
> Gradient-based metrics cannot be used to directly quantify memorisation, but often (efficiently) measure related quantities (e.g. sample difficulty).

## 5.2 Quantifying memorisation using information theory

The next set of techniques we discuss quantify the "flow of information" from data points to the parameters of a model using the tools of information theory. A subset of these works rely on the concept of Shannon mutual information (MI) (Kolmogorov, 1956), which quantifies the amount of "information" which can be deduced about a variable by monitoring another variable (i.e. the amount of "dependence" between the variables) (Shwartz-Ziv, 2022; Goldfeld et al., 2018). However, while such methods have strong theoretical foundations, they ultimately suffer when placed in the context of ML. This is because these approaches assume that all inputs of interest including the resulting ML model's weights are random variables, which is often not the case, as most ML approaches result only in a single set of weights. Moreover, Shannon information assumes that computational abilities are generally unbounded. This means that, in the view of Shannon information theory, information cannot be increased by further processing. This stands in contrast to real-world practice, in which computation is constrained by cost or by the available time, and information is thought to be "extracted" from data by the ML model.

This motivated the development of usable information theory and its variant of information termed $\mathcal{V}$-information (or $\mathcal{V}$-usable information) (Xu et al., 2020). This method can be seen as a computationally constrained version of Shannon MI, which measures the "usable" information contained in the data, which can be extracted by functions (e.g. ML models) in a specific family (denoted $\mathcal{V}$). To mitigate the aforementioned issue of non-randomness, Xu et al. (2020) proposed to use the entropy of a softmax function (which is in itself a distribution). This interpretation has been applied to quantify the difficulty of datasets (Ethayarajh et al., 2022) and to localise information leakage to specific components (layers) of ML models (Mo et al., 2021). The authors of Mo et al. (2021) also correlated $\mathcal{V}$-information with Jacobian sensitivity, which corresponds to the norm of the gradient with respect to the model's input, linking this study to the techniques discussed above. They deduce that there are two "types" of useful information the model can memorise: *input* and *latent*. The former represents the model's ability to correctly recall the input data under reconstruction attacks (Zhao et al., 2019), while the latter corresponds to information about properties of the data (which can be exposed through attribute inference attacks, discussed below).

Orthogonal to the domain of usable information theory are approaches which attempt to enable the computation of information-theoretic quantities in high-dimensional ML models. Notable among these is the technique of estimating e.g. MI through randomised low-dimensional projections, pioneered in Goldfeld & Greenewald (2021) and termed *sliced* MI. Sliced MI was used in Wongso et al. (2023) to measure memorisation and captures the notion of average usable information (as it averages the MI for all random projections) instead of the largest usable information. We contend that the relationship between the difficulty of learning,

the complexity of the representations at hand and the memorisation properties of the network are still a nascent area of research of high potential impact.

Another noteworthy approach was proposed by Harutyunyan et al. (2021) and is termed *smooth unique information.* This formulation considers the distinction between what data the weights of the model contain against what the model actually learns. Similarly to influence functions, authors consider how a single included/excluded instance can influence the training of the model and measure it using a smoothed KL-divergence between the two models (concretely: its expectation over the label distribution). Authors conclude that removing highly informative samples identified by their method results in a stronger performance degradation compared to removing the same quantity of uninformative/random samples instead. These findings again highlight the divide between studies on informativeness of individual samples and the extent to which they are prone to memorisation. While this work studies the effect of including or excluding highly informative (in a manner of speaking – influential) samples on generalisation performance, no analysis is performed in terms of the influence of training samples on themselves.

> **Key Point**
>
> Techniques from the domain of information theory can identify the samples which are likely to be memorised by considering the amount of information that these samples contain (and, hence, the information learnt by the model). These approaches can, however, suffer from unrealistic assumptions and poor computational performance.

## 5.3 Measuring sample difficulty

The concept of sample difficulty is often closely related to memorisation (e.g. "difficult" samples can be more prone to memorisation), but is arguably ill-defined. One can intuitively interpret a difficult sample to be one which is fit poorly by the model. However, the reasons for such poor performance can vary: the sample can contain "rare" features (Agarwal et al., 2022), have poor quality (Usynin et al., 2023b) or the sample can come from a data-generating distribution which is mismatched to the task at hand (Chen et al., 2021). A number of methods were proposed to identify at which point different "concepts" are learned from individual features of a sample, showing that those that correspond to "easier" concepts are identified much earlier in the model (Baldock et al., 2021). Alternatively, the authors of Garg & Roy (2023b) rely on the curvature of the loss function to assess how "clean" individual samples are (where higher cleanness signifies a stronger representative of a class). However, similarly to other works on example difficulty, the notions of "cleanness" are closely intertwined with samples being "typical" representatives, which can have different interpretations based on the input modality (i.e. defining a typical cat image vs. a typical job description). Notably, this method through an approximation of the Hessian trace produces consistent results across different architectures and random seeds (which was a limitation of similar approaches (Agarwal et al., 2022; Pruthi et al., 2020)).

Arguably, the best example of the challenges associated with the interpretation of "sample difficulty" is discussed in Carlini et al. (2019a), as this work employs five separate metrics in order to identify the connections between individual image characteristics (for instance, adversarial robustness on these images) and its "difficulty". While authors conclude that most of these metrics are correlated amongst each other, there is still no causal relationship between e.g. holdout accuracy and "difficulty" of a sample. Another work of Harutyunyan et al. (2021) shows that example difficulty can be estimated by observing the behaviour of the loss values on individual samples on their "corresponding" test-time points. What this definition proposes, is to disentangle concepts of train-time performance and example difficulty, and to, instead, consider related data points, that the model has not encountered during the learning process.

Finally, another noteworthy work which tries to approximate example difficulty was proposed by Ethayarajh et al. (2022) using $\mathcal{V}$-usable information (which is a computationally constrained version of Shannon entropy). What we are interested in here is the similarity of this definition to the notion of sample influence. Here authors estimate the difficulty of a sample as a model's predictive performance on two data distributions, one of which contains a record of interest and one does not. This metric describes the difficulty of an entire

dataset, given a model $\mathcal{V}$, but authors also proposed pointwise $\mathcal{V}$-information (PVI), which can be used to describe the difficulty of individual samples, similarly to the works we discussed above. One noteworthy conclusion of this work is that PVI, can serve as a context-agnostic measure of "difficulty" across different models and modalities if using it as a threshold (i.e the PVI value after which samples start to be misclassified and are hence considered to be "difficult"). These methods establish an indirect connection between difficulty and memorisation: Difficult samples can often be those that fall on the tails of the data distribution and are, hence, more prone to memorisation. However, reasoning over sample difficulty is not straightforward, because while difficult samples may be highly self-influential (e.g. mislabelled data), they do not necessarily have to be (e.g. poorly acquired images).

To summarise: it is speculated by several works that the identification of "difficult" samples is similar to the identification of samples which are likely to be memorised. These methods can even be perceived as proxies to the aforementioned influence functions using only first-order information. While the definition of "sample difficulty" has not been unified yet across various works, most of the formulations that we discussed base their definitions of the concepts of "atypicality", making detection of such samples a challenging, yet an important task across any input domain.

> **Key Point**
>
> Samples can be difficult for many reasons, but samples which are deemed to be more difficult are often memorised more.

Overall, we conclude that there exists a large number of formulations which are often described as "memorisation" of some form. We, however, highlight that while many of these measure quantities **related** to memorisation and identify samples which can often be more prone to memorisation, these **cannot** be used as direct measurements of this phenomenon.

## 6 Localisation and timing of memorisation

Two intuitive question which arises from the discussion above is:

1. Where does memorisation occur in the model?

2. When does memorisation occur during training?

To reason over the localisation of memorisation, we first need to consider the localisation of *learning* in ML models. One of the fundamental works presented by Bau et al. (2017) shows that individual concepts that are included in a data point (e.g. texture or color of an object on an image) are learnt by different parts of a neural network. Authors demonstrate that during the learning procedure, different parts of the model become "responsible" for learning individual concepts contained in the input. Thus, authors argue that it can be possible to identify individual neurons associated with specific concepts.

As learning of individual concepts can be localised to different parts of the model, it was previously established that this also holds for memorisation. It was shown in the works of Mo et al. (2021); Baldock et al. (2021); Maini et al. (2023) that as part of the learning process, memorisation can also be associated with specific parts of a model. Specifically, authors of Baldock et al. (2021) present *prediction depth*, which is the earliest layer in a model based on the representations of which it is possible to make a correct prediction on the input. Authors determined that "easier" concepts (i.e. those associated with generalisation) are learnt in the earlier layers, whereas the more complex ones are learnt in the last layers of the model. The work of Stephenson et al. (2021) further shows that memorisation of atypical (mostly mislabelled) samples can also be localised to the last layers of the model, where most mispredictions also occur in the early stages of training. It is of note that the work of Mo et al. (2021) shows a similar result through the lens of gradient information leakage, which we have previously discussed in Section 5. The fact that memorisation can be localised was further supported by the work of Maini et al. (2023). However, authors of Maini et al. (2023) show that contrary to the results of Baldock et al. (2021), memorisation, while localised to specific parts of

the model, is **not** associated with the final layers. They instead show that memorisation is indeed localised to specific neurons, but these are often distributed across multiple layers. Moreover, the authors of Wongso et al. (2023) conclude that learning patterns can be different based on the architecture and the depth of the model: convolutional models exhibit more "learning" in the deeper convolutional layers, similar to the discussion above. In turn, multi-layer perceptron-based models exhibit an increase in the amount of useful information the model learns approximately linearly. Additionally, Hernandez et al. (2022) previously shown that certain architecture-specific parts of the model may even be "responsible" for memorisation altogether (in this case the attention heads). These findings highlight that learning (and memorisation) is not only inconsistent across different model types (i.e. different models can extract different information), but also that even individual *layers* in a model learn differently from the same data based on how deep they are (i.e. learning and memorisation can be localised to specific parts of the model).

In regards to the question of timing, authors of Arpit et al. (2017) show that memorisation is prevalent at specific stages of training. Concretely the authors showed that models tend to start learning "simple" patterns first, thus showing that generalisation can often occur before memorisation (supported by Kishida & Nakayama (2019)). Additionally, the work of Paul et al. (2021) shows that it is also possible to determine the influential samples early on and use these to guide the training process, by removing samples of low influence from the training dataset. Therefore, it is also possible to not just link memorisation to the training phase, but to also employ the training data to force the model to change its memorisation pattern (i.e. which samples get memorised). This mirrors a finding by Baldock et al. (2021), who relate memorisation to *prediction depth*, i.e. the depth of the layer at which representations which effectively determine the network's prediction on a specific sample are formed. This highlights that a higher difficulty of forming a defining representation in a network is associated with a higher probability of memorisation. Another line of work has previously explored the issue of localisation of individual learning concepts (including memorisation and generalisation) relies on the tools from the domain of mechanistic interpretability (Pearce et al., 2023; Nanda et al., 2023; Hernandez et al., 2022). In Nanda et al. (2023) it was shown that using these tools it is possible to identify three distinct stages of ML training: memorisation, circuit formation (i.e. generalisation) and clean-up (i.e. replacing the memorised information with generalised patterns). This work contradicts the findings of Arpit et al. (2017), arguing that memorisation occurs prior to generalisation, showing that while memorisation *can* often be temporarily localised, it is not yet possible to concretely determine when it takes place.

Overall, although there is no clear consensus regarding the specifics, evidence from multiple prior works supports that memorisation is a process that can be localised both spatially and temporally. We note, however, that while the aforementioned works have made significant progress in this field, there is, yet, no clear consensus on how and where to attribute individual learning concepts (and, hence, memorisation) to.

> **Key Point**
>
> Memorisation is a process that can be localised both spatially and temporally (e.g. at a specific point of training or the layer of the model).

## 7  Inducing deliberate memorisation

For many learning settings and in particular, for generative models, it is often very difficult to conceptualise what the model memorises (i.e. does producing an output "similar" to the training record count as memorisation?) One intuitive method of evaluating this involves deliberately inserting data samples which are "expected" to be memorised by the model due to their "atypicality". The capacity for memorisation is then measured with respect to how well the model can reproduce these inputs as its output (i.e. for generative models) or how performance on these data points compares to performance on the rest of the dataset (e.g. for supervised learning).

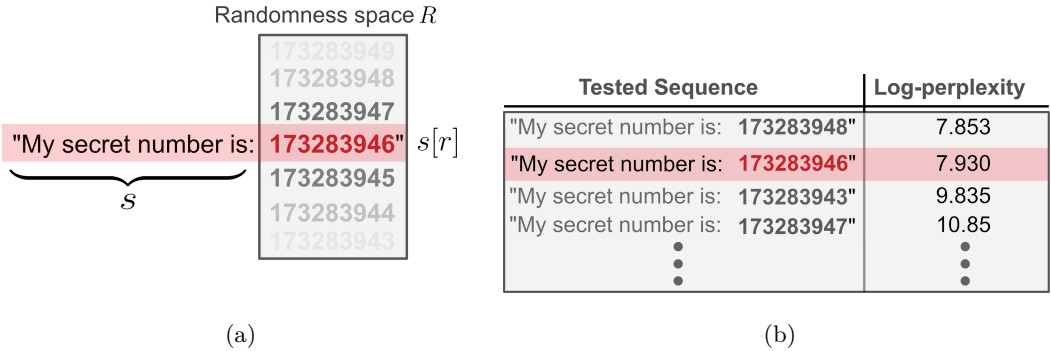

(a)               (b)

Figure 2: Example workflow of measuring memorisation through an inserted canary (shown in red). First, a canary is generated by combining some pre-defined structure $s$ and some randomness $r$ drawn from some randomness space $R$ (e.g all possible nine-digit numbers) (a). After inserting the canary into the training dataset, the ability of the model to reproduce the canary is computed as its *exposure*. This measure is determined by $|R|$ (the size of $R$) and the canary's index (rank) in table (b), ordered by log-perplexity and containing all other possible canaries from $R$ (that could have been inserted but were not).

### 7.1 Canary memorisation

The, arguably, most widely used approach for inducing (and empirically quantifying) memorisation was proposed by Carlini et al. (2019b) and has since been employed in many machine learning contexts (Thakkar et al., 2020; Carlini et al., 2021; Tirumala et al., 2022; Lee et al., 2021), particularly in the field of language processing. In essence, the authors propose to measure how much a generative language model can memorise by purposefully inserting crafted data samples (canaries) into the training dataset. These crafted samples are designed to require memorisation. The connection to the long-tail definition of Feldman (2020) is that such canaries are constructed to resemble data from low-density regions of the training data distribution. These samples are (often informally) called anomalous, outliers or atypical. Note that in supervised learning, a similar effect can be achieved through deliberately mislabelling the example. The "exposure" of the canaries is measured and used as a proxy for the memorisation of other atypical samples.

To define exposure, we first discuss the notion of *log-perplexity* it is based on. Intuitively, log-perplexity measures how "expected" a given sample is (or how much the model is "surprised" by this sample) and is defined as follows:

$$\text{perp}(x, f) = - \log_2 p(x_1, ..., x_m | f) \tag{6}$$
$$= \sum_{i=2}^{m} (- \log_2 p(x_i | f(x_1, ... x_{i-1})))$$

where $f$ is the model, and $x$ is the input sequence of length $m$. Here and below, we use $p$ in a slight abuse of notation to denote both probabilities and likelihoods.

To generate a canary ($s[r]$), we combine a predefined, fixed structure $s$ and some randomness $r$ drawn from some predefined randomness space $R$ (e.g. the space of nine-digit numbers, see Fig. 2a). This generated canary is then inserted into the training dataset. After model training, we compare the perplexity of our inserted canary to all other possible canaries that we could have inserted. Perplexity, however, is a relative measure and highly dependent on the specific training setup, including model architecture, dataset composition and the application. Thus, Carlini et al. (2019b) proposed a "relative" notion of perplexity called *rank*, i.e. the index of a canary in the relative ordering of perplexities across all possible canaries (see Fig. 2b).

Exposure, in turn, is then simply determined by the size of the chosen randomness space and the canary's rank. Formally it is defined as follows:

$$\text{exposure}_f(s[r]) = \log_2 |R| - \log_2 \text{rank}_f(s[r]), \tag{7}$$

where $|R|$ is the size of the randomness space from which the canary $s[r]$ was generated, thus making exposure a positive value.

The exposure definition has been successfully transferred to other machine learning modalities, such as computer vision in Hartley & Tsaftaris (2022), but due to the complexity of canary generation in other domains, most follow-up works in the area concentrated on language models (LMs) instead (Thakkar et al., 2020; Carlini et al., 2021; Tirumala et al., 2022). However, canary-based memorisation definitions are significantly less generalisable than the one discussed in Section 3, as they (1) require the data owner to generate canaries (which are specific to their setting) and (2) are particularly well-suited to language settings, compared to other generative modelling tasks.

Overall, canary-based approaches have seen popularity due to their simplicity of implementation and high applicability to generative ML tasks (Liang et al., 2023; Nijkamp et al., 2022; Taylor et al., 2022; Alayrac et al., 2022; Wei et al., 2022). One important finding that authors of Carlini et al. (2019b; 2022b; 2023)) report, is that exposure seems to increase drastically with the number of times a canary is seen during training. The authors thus suggest that if an atypical sample is included many times in the training dataset, it is much more likely to be memorised. This is in contrast to follow-up work Zhang et al. (2021a), which has argued that such a conclusion is ill-guided, as extraction/generation-based memorisation quantification methods (like canaries) are biased towards identifying frequently occurring samples (Lee et al., 2021; Zhang et al., 2021a). In fact, Zhang et al. (2021a) presents contrary evidence to the prior claim: using re-training-based influence estimators (from Section 4.1) the authors show that memorisation estimates tend to actually decrease for data points that are repeated many times. This finding thus raises the question of the validity of memorisation and privacy risk conclusions drawn from extraction-based memorisation experiments, as the memorisation of frequently occurring sentences (or data points) potentially poses little privacy risk compared to rare sentences. This is because frequently occurring sentences such as commonly known facts typically pose little privacy concern compared to e.g. personally identifiable information which only occurs once.

> **Key Point**
>
> Unintended memorisation can, in some cases, be measured by inserting samples into the training dataset which are crafted to be more prone to memorisation and measuring how likely they are to be regurgitated by the model.

## 7.2 Memorisation and adversarial samples

The aforementioned canaries are deliberately crafted to resemble samples from the low-density region of the data distribution. They are, however, still valid (albeit atypical) data points and can be used to train a well-generalised model. The same concepts are exploited in works on data poisoning, where a malicious actor artificially generates data points from the low-density region of the distribution, which are crafted to degrade the performance of the model. These data points produced during this process are known as adversarial samples (Goodfellow et al., 2014) and these were originally designed to showcase how tiny invisible input perturbations can be used to cause an image classification model to mispredict, but were later adapted to other settings and modalities (Tian et al., 2022). These, unlike canaries, are applicable to both the discriminative and the generative settings (and are assumed to have high negative cross-influence in addition to high self-influence).

In many cases, a very small proportion of such samples ($< 1\%$) suffices to severely harm the utility of the trained model (Chobola et al., 2023; Zhou et al., 2021). Adversarial data points are generated to be atypical through either incorrect labelling, embedding of features which are associated with a different class, addition of imperceptible noise etc. (Usynin et al., 2021). As a result, these samples can often be used to manipulate the behaviour of the model as these are A) more likely to be memorised and B) only require

a small perturbation, making these attacks difficult to detect. Adversarial samples can also be used to aid attacks on ML models, aiming to extract the information that the model has memorised (particularly for underrepresented samples, which were shown to be more vulnerable in Shokri et al. (2017)) about individual data points, which it would not expose otherwise (Tramèr et al., 2022; Carlini et al., 2022c; Chobola et al., 2023; Bagdasaryan & Shmatikov, 2021). For a more in-depth discussion on how these adversarial data can be generated in different learning settings, we refer to Tian et al. (2022).

> **Key Point**
>
> Adversarial samples are made to be artificially atypical and hence have a higher influence on the model (malicious or otherwise).

## 8 Privacy implications of memorisation

Memorisation is not a process that occurs in a vacuum: it likely affects the privacy of individuals, whose data is used to train a ML model. To assess the extent to which this phenomenon can harm these individuals, we discuss the implications of memorisation through the lens of data privacy. We note, that while our work is primarily aimed at centralised ML settings, we additionally include a detailed discussion on memorisation in decentralised settings in the Appendix. We are particularly interested in answering these questions over the course of this discussion:

- What are the implications of memorisation on privacy of the individuals and which definition(s) can we use to estimate these?

- What methods exist to extract the information memorised by a model?

- How can memorisation affect privacy of generative models?

### 8.1 Memorisation and privacy attacks

Models which have memorised much of the data they have been trained on can often be perceived as more "dangerous", as in many cases it is possible to extract the information that has been memorised. While attacks can be executed in both the discriminative and the generative settings, we are particularly interested in discriminative models. For these extraction of memorised information often takes the form of attacks on privacy, as they are not designed to output the data they were trained on. We discuss which "individual" traits of ML training can be exploited to extract the information the model memorised about individual samples (similarly to how many of **the same** metrics can be used to identify influential or atypical samples). Concretely, we discuss three major privacy attack methods, namely: membership inference, attribute inference and data reconstruction (often referred to as data inference or model inversion (Usynin et al., 2021)).

**Inference attacks** One of the more prominent privacy attacks is termed a *membership inference attack* (MIA) (Shokri et al., 2017) and it identifies if the model's training dataset contains a specific point of interest. This attack has a flexible threat model and can be employed against shared white-box models (Sablayrolles et al., 2019) or API-only black-box models (Choquette-Choo et al., 2021) with high effectiveness. Similarly, MIAs can successfully target both the discriminative and the generative models (Chen et al., 2020a; Liu et al., 2019a; Hilprecht et al., 2019), making them a versatile context-agnostic "unintended memorisation" auditing tool. There is growing evidence to support that high memorisation is associated with an increased MIA vulnerability, although it seems that memorisation is not necessary **and** sufficient but just sufficient (i.e. where memorisation leads to being vulnerable to MIAs) and that there are other reasons for vulnerability beyond memorisation (Choi et al., 2023; Carlini et al., 2022c).

It was previously shown that MIAs are particularly effective against models with larger generalisation gaps (Shokri et al., 2017; Usynin et al., 2022; Truex et al., 2019; Salem et al., 2018) (with more examples in Dionysiou & Athanasopoulos (2023)). For instance, a suite of attacks all of which rely on comparing the loss

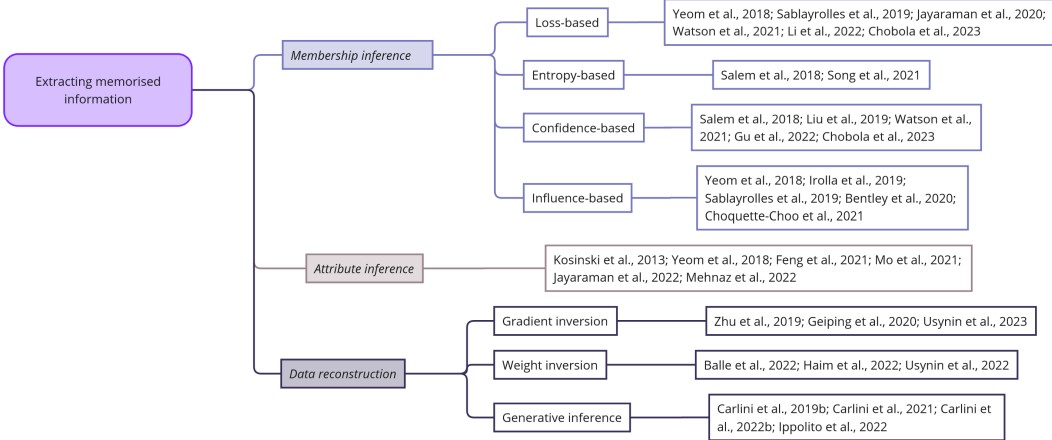

Figure 3: Brief overview of various attacks on privacy and the information they exploit.

values on previously seen and unseen samples (Chobola et al., 2023; Carlini et al., 2022a; Yeom et al., 2018; Jayaraman et al., 2020) naturally benefit from lower loss magnitudes on the data the model was trained on. However, we note that while attacks using model loss, outperform the naive prediction-based approaches (e.g. (Yeom et al., 2018; Choquette-Choo et al., 2021; Sablayrolles et al., 2019; Irolla & Châtel, 2019; Bentley et al., 2020)), they typically represent a non-membership test (i.e. are mostly effective at determining who was **not** a member rather than who was one) (Li et al., 2022). Similarly, any proxy metric, such as the confidence scores, can be used to identify whether a training record was previously "seen" by the model (Shokri et al., 2017; Salem et al., 2018; Watson et al., 2021; Gu et al., 2022). However, it was previously shown in Yeom et al. (2018) that overfitting is **not** a prerequisite for a successful attack. Moreover, when one considers attacks on generative models (e.g. those described in Chen et al. (2020a); Hayes et al. (2017); Hilprecht et al. (2019)), it is not clear what overfitting even implies.

Works such as Jayaraman & Evans (2019); Song & Mittal (2021); Cohen & Giryes (2024) attempt to make a more concrete connection between how memorisation of data points affects their inference attack susceptibility (based on the notion of memorisation presented in Section 7.1). This formulation permits a direct evaluation of how well individual samples can be A) memorised and B) inferred via MIA. A number of recent works, such as Cohen & Giryes (2024) and Carlini et al. (2022a) have speculated that there is a correlation between samples which are more likely to be memorised and the samples which are more vulnerable to adversarial inference (and MIAs in particular). Moreover, results from Choi et al. (2023) suggest that this is indeed a phenomenon which is applicable to memorised samples only, and not to "atypical, OOD" or "rare" samples. This is primarily because the definition of OOD is specific to an individual learning context and depends on the modality, which features are considered to be OOD etc. Additionally, works such as Cohen & Giryes (2024); Usynin et al. (2022); Leino & Fredrikson (2020) show that techniques which reduce the amount of information a model can memorise about individual samples (such as differential privacy (Dwork, 2006)) have been used to effectively mitigate MIAs in the past.

So while a direct causal relationship between memorisation and susceptibility to MIA has not yet been established (and although it seems likely), there are still two noteworthy observations. Firstly, atypical samples which are more likely to be memorised, are also more susceptible to inference attacks, showing that there is a connection between how much can be memorised by the model and how much can then be "exposed" by the model (Mo et al., 2021; Cohen & Giryes, 2024; Choi et al., 2023). Secondly, similarly to the formulations of memorisation, the extent of MIA's success can also depend on memorisation and generalisation simultaneously, showing that these notions are complementary.

Another attack, which allows the adversary to determine individual features or attributes of their victim is termed as *attribute inference* (Kosinski et al., 2013; Jayaraman & Evans, 2022; Mehnaz et al., 2022). For instance, it was previously shown that attribute inference attackers are capable of inferring protected demographic information from parameter updates alone (Feng et al., 2021; Jourdan et al., 2021). As previously

discussed in Mo et al. (2021), the risk of this attack can directly depend on the amount of information that is "embedded" in the trained models (more so in deeper layers (Maini et al., 2023)). Therefore works such as Thomas et al. (2020); Jourdan et al. (2021) hypothesise that similarly to MIAs, this attack also benefits from memorisation of individual samples and is particularly effective against data points of high influence.

**Model inversion** The last type of privacy attacks that we discuss is model inversion attacks, which aim to extract the training data given some representation of the model (e.g. consecutive training snapshots (Usynin et al., 2022), final model (Haim et al., 2022), gradient updates (Zhu et al., 2019; Geiping et al., 2020; Usynin et al., 2023a)). The unique trait of these attacks is the fact that they can be used against *different* representations of the same model in order to extract the training data.

When quantifying memorisation under these attacks it is important to clearly disentangle the notion of memorisation and "leakage" (or exposure). Memorisation is the amount of information the model can *store* about individual (usually rare) samples, whereas leakage is the amount of information that the model can *reproduce* when queried. Therefore, while many of these attacks show "how much models leak", these leakages can often A) represent the information about a class of input samples rather than individual samples (e.g. reconstruction in Fredrikson et al. (2015)) and B) not capture all of the information memorised by the model (e.g. when extracting an image, a lot of pixel variation is possible, making it unclear what exactly was memorised by the model, while still showing "some" sensitive attributes (Usynin et al., 2023a)). These attacks typically exploit phenomena that are similar, but otherwise disjoint from memorisation (discussed in Section 5). Notions such as $\mathcal{V}$-information (Mo et al., 2021) can be employed to identify the amount of useful data contained in the shared gradient, while a more direct metric (e.g. canary-based approaches (Carlini et al., 2019b)) could be used to quantify the amount of memorisation occurring over training that gets exploited by an inversion attacker.

The aforementioned attacks (in particular MIAs) can be used to "audit" the model and to empirically identify information it memorised. This was previously manifested in A) works which verify is a specific training sample was memorised during model training (Jayaraman & Evans, 2019; Song & Mittal, 2021) and B) works validating the bounds on how much memorisation can occur with respect to specific samples (Ye et al., 2022; Nasr et al., 2021). The former allows privacy-conscious individuals to identify if their data was likely used (and memorised) by the model during training. The latter can help both the data and the model owners to establish how much information can be both memorised and exposed once a trained model is made public (discussed in detail in Section 9.3) to introduce more realistic privacy bounds.

Finally, it is important to discuss how memorisation (as well as attack susceptibility) of one record can directly affect another record. We have previously established that data points which are memorised more (and have a higher influence) are typically associated with being more susceptible to privacy attacks. Thus, intuitively we may deduce that removing these records would reduce the amount of memorisation a model can experience and, hence, make the entire settings more robust against privacy attacks. Carlini et al. (2022c) discovered that this, in fact, does not seem to be the case. They termed this phenomenon as *the privacy onion effect*, which states that should a sensitive record be removed from the training dataset, the model can relatively quickly start memorising other records more to fill the information that becomes missing upon an exclusion of the record of interest. Hence, authors give an "onion" analogy: when the top privacy-sensitive layer of data points gets removed (i.e. records which are most susceptible to privacy attacks), the layer that follows is becomes more exposed and, thus, becomes more susceptible to the same attacks. This phenomenon directly links to the rest of our work: the amount of memorisation that the model experiences with respect to a given data point is **relative** i.e. all other things being equal, the rest of the data used to train this model (and in particular its distribution) can affect how much model memorises with respect to this point. It is also, to the best of our knowledge, one of the few works to identify a direct relationship between exposure, attack susceptibility and memorisation, further proving that all of these are distinct, yet complimentary processes. Note also that –by design– auditing measures *leakage* from the model and not memorisation; it thus stands to reason that auditing surfaces the points on the "outer layers of the onion", i.e. the most susceptible points, rather than the most memorised ones.

Overall, identification of the risks associated with model memorisation is a non-trivial task, which relies on a large number of methods. More importantly as underrepresented populations are much more likely

to be memorised (Feldman & Zhang, 2020) and are often more susceptible to attacks (Kulynych et al., 2019), causal reasoning over the privacy risks associated with memorisation is an important consideration in development of secure AI systems.

> **Key Point**
>
> Data points which get memorised more are also often more vulnerable to most privacy attacks.

### 8.2  On memorisation in generative models

As generative ML becomes more prevalent (Liang et al., 2023; Nijkamp et al., 2022; Taylor et al., 2022; Alayrac et al., 2022; Wei et al., 2022), we find that the existing formulations of generative memorisation (and even more so - generalisation) have very different privacy implications compared to the discriminative settings. The question which naturally arises when discussing generative models is: as new data is generated based on the previously seen one, can the model compromise privacy of an individual by generating data close to the one they shared during training? This question becomes even more challenging to answer, as the community often distinguishes between the "intended" (i.e. the one that contributes towards generation of useful facts) and "unintended" (i.e. the one that reveals PII during generation) memorisation, making it difficult to identify which type of memorisation is a given definition capturing (Naseh et al., 2023; Carlini et al., 2021). There exists a large number of prior studies on the privacy of generative models (Liu et al., 2019b; Triastcyn & Faltings, 2020; Wu et al., 2019; Carlini et al., 2021; 2019b), one of which we have previously discussed in detail in Section 7.1. The main takeaway from this discussion is that for individuals (and their data) which fall on the tails of the data distribution (i.e. of high influence), generative models can present a greater privacy concern, as their data is more likely to be memorised (particularly in language models) (Carlini et al., 2019b; 2021; Ippolito et al.). Moreover, given the recent advances in the field of LLMs with many models exceeding billions of trainable parameters (Touvron et al., 2023; Jiang et al., 2024), it was shown that as the model's size (and its effective capacity) grows, so does the capacity for memorisation (Carlini et al., 2019b). While no *formal* relationship between model's effective capacity and its capacity to memorise has been established, most prior evidence shows that these are, in fact, positively correlated, presenting additional privacy challenges when training large generative models on sensitive data (Carlini et al., 2022b; 2019b; Meehan et al., 2024). Some solutions to this problem mostly rely on training data sanitation (e.g. the removal of personally-identifying information (PII) (Inan et al., 2021) or deduplication of the training data (Carlini et al., 2022b)), as well as DP training (Zanella-Béguelin et al., 2020; Shi et al., 2022; Du et al., 2023). Both approaches have proven to be effective at reducing the risks of unintended data exposure, but the former requires access to the entire training dataset and the latter can often result in poor model utility.

The question which remains unanswered concerns the metrics which can be used to measure this unintended memorisation. While it is sometimes possible to generate the **exact** training data used to train language models (e.g. in Fig. 2), how could one measure the same notion in a domain, where such one-to-one mapping is unlikely and where data can be generated with "similar enough" private features corresponding to the training data? Moreover, authors of Ippolito et al. and Lee et al. (2021) argue that even for LLMs, there is no straightforward way to describe memorisation of sensitive data (as the models are capable of producing outputs, which are syntactically different, but semantically identical, making "verbatim memorisation" a poor privacy metric). Similar conclusions were presented by Carlini et al. (2023) for images and it was termed as the *eidetic* memorisation (closely linked to the concept of "episodic" memorisation from (Zhang et al., 2021a)), which refers to model's ability to output an image it has only seen once (or $k$ times for a $k$-eidetic memorisation, where $k$ is a small integer). In order to quantify memorisation in this domain, authors proposed to use a threshold $\delta$, which measures the distance between an image and similarly-looking neighbours, alleviating the need to rely on an exact image reconstruction.

However, one may argue that since pixels (on their own) are not necessarily meaningful (i.e. change in a single pixel is unlikely to affect the semantic meaning of the image), but the features within natural images are, without further contextualisation most pixel-based metrics are meaningless. The work of Fernandez et al. (2023) tries to address this issue by using a similarity metric (proposed in Packhäuser et al. (2022))

based on the re-identification ratio: can the adversary identify the private data of a specific individual based on the image generated by the adversary? Another method was previously proposed by Kuppa et al. (2021) and here authors measure generative sample memorisation through susceptibility of individuals whose data is used to train the model to privacy attacks, in particular membership inference. We discuss how memorisation is related to information leakage in Section 8.1 in detail, but as we can already see it is difficult to disentangle the two and measure the degree of memorisation directly. Therefore, we conclude that while in discriminative settings there exists a number of methods which aim to identify specific features or samples the model "memorises", in the context of generative modelling, these are challenging to contextualise.

> **Key Point**
>
> Memorisation in generative models can be perceived either through self-influence or as the ability of a model to produce outputs which are similar to the input data.

## 9 Preventing and reversing memorisation

As information can be memorised by the model, it raises the question whether it can also be "unmemorised". This process can occur naturally over the training process and it can be induced manually by the data owner. Moreover, over the course of training it is also possible to bound the amount of information that the model can memorise through the use of DP training. In this section we discuss these phenomena and outline the processes which are responsible for reversal or reduction of memorisation.

### 9.1 Spontaneous reversal of memorisation

While some data points are more prone to memorisation, the information contained in the model about these memorised samples often changes over time (due to model's limited memorisation capacity). This effect is often termed as "forgetting", which intuitively means that if the model does not encounter a previously seen training point during training (i.e. if it "disappears" from the training set), the performance of that model on this specific training point would degrade over time (Toneva et al., 2018). The main question to consider is: "is forgetting the opposite of memorisation?" Intuitively, the answer is affirmative, as when the model forgets the data point, model's performance on it degrades. This effect can be particularly profound for the data points which fall on the tails of the distributions (e.g. mislabelled data points), with Jagielski et al. (2022) showing that data which is more likely to be memorised under Feldman & Zhang (2020) definition is also more likely to be forgotten.

It was previously shown by Feldman (2020) that models *have to* memorise certain information about individual samples over the course of model training. Authors of Jagielski et al. (2022) went further and argued that similarly to how models *have to* memorise, they also *have to* forget information about individual samples if they are not encountered again during later stages of training. As deep learning often requires multiple sources of non-determinism to be present (e.g. random batch shuffling, stochastic gradient methods etc.), these are, therefore causing the models to forget. Authors additionally demonstrate that should non-determinism be eliminated in full, forgetting no longer occurs, which is "in spirit" similar to the conclusions of Feldman & Zhang (2020). In Feldman & Zhang (2020) authors formalise that if a source of non-determinism is included during training (e.g. addition of noise under differential privacy), then the models would be limited in the amount of information they can memorise (and, hence, forget).

The work of Jagielski et al. (2022), again, shows that the samples which are most vulnerable to this phenomenon are the atypical samples. This notion has different implications to "catastrophic forgetting": where the entire sub-distribution of data (e.g. an entire class) can be forgotten, which is not guaranteed to directly affect the previously memorised samples (but rather some population of the training set, which *could have been* memorised). This phenomenon can be measured using the concept of adversarial advantage, which compares the performance of membership-based attacks on the samples of interest at different stages of training. Moreover, the issue of forgetting captures another important question that we have previously outlined: when "forgetting", one is required to define what is that needs to be "forgotten" (i.e. what has previously

been memorised)? In Tirumala et al. (2022), for instance, this concept captures verbatim memorisation, which authors of Jagielski et al. (2022) argue does not represent a more general definition of memorisation. Similarly to other works on memorisation in language models, they craft uniquely identifiable samples (i.e. canaries from Carlini et al. (2019b)) to approximate the degree of memorisation that remains on specific training records over the course of training.

> **Key Point**
>
> Samples that are memorised during training can be forgotten if they are not encountered again. This is particularly common for atypical data points.

## 9.2 Induced reversal of memorisation

Finally, we briefly mention another area which is closely linked to both notions of memorisation and forgetting: machine unlearning (Bourtoule et al., 2021). Unlearning is a set of different techniques, which given a training record(s) modifies the model to no longer include the contributions of that record(s). It is of note that forgetting *can* be targeted as presented in Zhou et al. (2022), showing that these phenomena are related. The difficulty of unlearning depends on a variety of factors, including the current state of the model (i.e. if it is currently training or if the model has already been trained), the features of the data record(s) as well as the remaining data points. Forgetting (when compared to unlearning) occurs naturally during training (it is often not possible to predict which specific records will be forgotten (Toneva et al., 2018)), whereas unlearning is a set of techniques designed to remove the contributions of specific data point(s). The final factor we outlined bears additional discussion.

As above, when performing unlearning, one tries to disentangle and remove the contribution of one or more data record. If this data point is "simple" or its class is over-represented, then the removal of its contribution is unlikely to have a large effect on the rest of the population (e.g. their attack susceptibility). However, if the record(s) comes from the tail end of the data distribution, its unlearning can negatively affect privacy of those, whose data remains in the dataset. We have previously discussed this phenomenon, namely the "the privacy onion effect" (Carlini et al., 2022c), which shows that by deliberately removing the record (and, thus, its contribution) that was more susceptible to attacks, other samples can become more prone to inference. Thus, we, again, observe that memorisation (and un-memorisation) is indeed relative to the data surrounding a highly memorised sample. As a result, this opens up a number of questions, such as "Does un-memorising also mean un-generalising? If yes, is it ethical to withdraw ones data if this will deteriorate the generalisation performance on other samples?". These questions must to be deliberated by the ML community in order to devise a unified strategy for effective model training.

> **Key Point**
>
> Forgetting can be artificially induced to "unmemorise" individual data points.

## 9.3 Bounding memorisation with differential privacy

Differential privacy (Dwork & Roth, 2014), as discussed previously, is the canonical privacy definition for statistical data processing and ML. Intuitively, DP can be perceived as a property of an algorithm, making it approximately invariant to an inclusion/exclusion of a single data point. More formally: The result of a computation (e.g. model training) is said to be *differentially private* if a probabilistic stability notion over neighbouring datasets is satisfied. Concretely, in the context of ML (and previously introduced notation), this stability notion requires that a specific model $f'$ (or set of weights) has a similar likelihood under $A(S)$ and $A(S\backslash i)$. This means that the value the random variable $f$ takes, cannot change much no matter which data point we remove from the training dataset. More formally, random Algorithm $A$ satisfies $\varepsilon$-DP when:

$$p\left(f'|A(S)\right) \leq e^{\varepsilon}p\left(f'|A(S\backslash i)\right), \tag{8}$$

for any $i \in S$ and where $A(\cdot)$ can take any value in some hypothesis class $\mathcal{H}$, given by e.g. the model architecture. Since the exclusion of any data point cannot significantly change the value $f$ takes, the output of $f$ on any sample $i \in S$ can also not be significantly impacted. This implies that changes in most metrics, due to the removal of a sample, (e.g. Eq. (3) or Eq. (4)) will be small and thus memorisation will be bounded under DP.

Concrete bounds on the memorisation ability (self-influence) of a differentially private model were previously presented in Feldman (2020) and van den Burg & Williams (2021). This is also directly related to the concept of model's *stability* described in Evgeniou et al. (2004), which is immediate from the results of Feldman (2020) and Brown et al. (2021). In particular, the former studies the relationship between the leave-one-out stability of an individual sample and model's memorisation capacity, showing that under DP, stability increases and memorisation decreases. Thus, informally, stable algorithms memorise less and DP algorithms are "very stable", leading to reduced memorisation. Further, using different interpretations of DP, it was also been shown that DP directly bounds the ability of an adversary to perform a MIA (Wasserman & Zhou, 2010) or training data reconstruction (Hayes et al., 2023; Stock et al., 2022). It is known that any privacy protection scheme which does not allow for catastrophic privacy degradation incurs some utility penalty (Dinur & Nissim, 2003). While applying DP in the context of ML also results in utility penalty (which may be unavoidable due to the fact that DP limits memorisation, as shown in Feldman & Zhang (2020); Wang et al. (2024)), recent works have demonstrated that, for many real-world datasets, competitive performance can still be achieved by applying a number of training adaptations such as pre-training on large public datasets (De et al., 2022; Berrada et al., 2023). Overall, there is strong evidence that DP is the tool of choice for bounding the negative privacy implications and risks of memorisation in practice.

> **Key Point**
>
> Differentially private ML training can (provably) prevent the negative traits caused by memorisation.

## 10 Recommendations for ML practitioners

Memorisation itself is a context-agnostic phenomenon which can be present in many ML tasks. However, the impact of memorisation can heavily depend upon individual training settings. Here we provide a number of guidelines, which should help ML practitioners contextualise memorisation in different learning scenarios and adapt their training settings with these in mind:

- Care should be taken when quantifying the *impact* of memorisation in different modalities. In language processing tasks, one should not measure memorisation verbatim (Jagielski et al., 2022; Lee et al., 2021; Ippolito et al.) and similarly, per-pixel metrics should not be used to quantify memorisation in imaging settings (Fernandez et al., 2023; Packhäuser et al., 2022).

- One should be careful to distinguish between memorisation, extractability and leakage. While these are all related phenomena, memorisation of a data point is not a pre-requisite for its successful extraction (Carlini et al., 2021) nor does it affect how much the model can leak about individual data samples (Ippolito et al.). Therefore, these terms should not be used interchangeably.

- Not all features are equally sensitive (e.g. PIIs vs common facts) or equally prone to memorisation (Carlini et al., 2022b; Inan et al., 2021). One can reduce the effectiveness of attacks exploiting unintended memorisation by curating one's training data (e.g. PII stripping) (Inan et al., 2021).

- Repetition of data can lead to its higher extractability (Jagielski et al., 2022; Tirumala et al., 2022). Therefore, deduplication of sensitive data can reduce the corresponding privacy risks.

- Generative models can be prone to regurgitation of the training data (Carlini et al., 2019b). This effect is more profound in longer context windows and larger models (Tirumala et al., 2022). Therefore, one should consider using smaller models and context windows, while maintaining an adequate level of utility.

- Underrepresented populations are more prone to memorisation (and to privacy attacks) (Mueller et al., 2022; Feldman & Zhang, 2020; Carlini et al., 2022b). Therefore, suitable protection techniques should be used when training models on such data.

- Removal of data from the training pool can have an adverse effect on the rest of the population with respect to privacy risks (Carlini et al., 2022c). Therefore, it is not recommended to remove individual data points which are perceived to be more memorisable and one should address these issues on a dataset or a model level instead (e.g. PII stripping) (Inan et al., 2021).

- Memorisation can be provably bounded by performing differentially private training, reducing the privacy risks and permitting public releases of models trained on private data (Feldman, 2020; Thakkar et al., 2020; Carlini et al., 2022c).

## 11 Conclusions and future directions

In this work, we systematically summarise and discuss the existing formulations of memorisation in machine learning models. We formulate the most widely used definitions and study their implications on the privacy of the data used to train the model. As quantifying memorisation using these definitions can be infeasible for many ML settings, we additionally discuss methods which allow the practitioners to estimate memorisation instead. Furthermore, we discuss methods which allow the data owners to identify which specific samples are likely to be memorised using techniques from the domains of data valuation and influence estimation. Additionally we provide a number of guidelines for ML practitioners, to help them to contextualise memorisation when designing ML workflows, allowing for a more safe and responsible AI training. Finally, we outline the open challenges and questions to be collectively addressed by the machine learning community in order to be able to standardise the taxonomy in the field and permit more well-designed collaboration with privacy risks associated with model memorisation in mind. Given a large number of metrics related to measuring memorisation, the effects memorisation can have on both the model performance and the privacy of the individuals, whose data is used to train the model, we outline a number of key future directions from which, we believe, the community can benefit:

- **Standardisation of the notion**: Many different definitions of memorisation are employed across various ML research fields, unifying these is of uttmost importance, as the misuse of the term can have implications in a broader community.

- **Information content and memorisation**: Samples with higher information content tend to be more prone to memorisation. However, as we established, not all information contained in those samples is useful to the model. Moreover, samples which are described as "more difficult" can be either highly informative (but rare) or of poor utility because they are malformed. While both of these sample types have high positive self-influence (i.e. if included, their utility on themselves improves), the former tend to be of high positive cross-influence, (i.e. improving the utility on other samples) and the latter are often of high negative cross-influence, harming the overall performance on other data points. We believe that further work is required to be able to establish strong links between information content, memorisation and the resulting model utility.

- **Privacy implications of memorisation**: Are some models or training settings more prone to "unintended" memorisation often exploited by attacks on ML? The community needs to establish clear guidelines on the implications of memorisation on the adversarial perceptibility, particularly with respect to potential privacy violations when generative models are adversarially prompted.

- **Memorisation in the age of AI regulation**: How can (and should) memorisation be regulated with respect to copyrighted materials frequently used to train large-scale models? Copyrighted material can often be accessed publicly, thereby allowing models to learn from it and memorise some of its contents. Since there is still no clear prior definition of what memorisation is, regulating how these should be processed is an open challenge.

- **Malicious memorisation and model alignment**: As we have previously seen, adversarial samples can present real threats to models trained on public data and are very frequently well-memorised by ML models. As a result, the alignment of such models can be severely affected by how well they learn (and memorise) from benign/malicious data, making it a promising area of future research (with prior works such as Zou et al. (2023); Carlini et al. (2024) suggesting that even in-prompt tuning can often be sufficient to override the existing alignment safeguards).

## Acknowledgements and Disclosure of Funding

This wark was supported by the German Federal Ministry of Education and Research and the Bavarian State Ministry for Science and the Arts under the Munich Centre for Machine Learning (MCML), the German Ministry of Education and Research and the the Medical Informatics Initiative as part of the PrivateAIM Project, the Joint Academy of Doctoral Studies of Imperial College London and the Technical University of Munich, the Bavarian Collaborative Research Project PRIPREKI of the Free State of Bavaria Funding Programme "Artificial Intelligence – Data Science", and the German Academic Exchange Service (DAAD) under the Kondrad Zuse School of Excellence for Reliable AI (RelAI).

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

# A  Appendix: Memorisation in decentralised settings

While there are no "fundamental" differences between memorisation in centralised and decentralised ML (as most decentralised ML frameworks can be seen as a large number of local centralised settings) there are two factors to take into consideration when analysing the issue of memorisation in these settings. Firstly, as participating agents (or clients) are often chosen at random, it is possible that sampling bias can severely affect both the training dynamics and the memorisation patterns. Secondly, local models need to be aggregated once local training is over, potentially affecting the memorisation that has previously occurred at a local level. In this discussion we employ a commonly used federated learning (FL) (Konečnỳ et al., 2016) framework for decentralised ML training. In FL it is often the case (particularly for cross-device FL, where only a handful of sites participates in joined training) that data is not independently and identically distributed (non-IID). This, in turn, means that many local models only have access to small datasets, often with highly personalised data points. In FL, each round the central server typically selects a random subset of clients. As it is impossible for the server to observe the local distribution of data at each client, it can be challenging to identify if the submitted update corresponds to a "typical" representative of the population or if it contains a large number of atypical data points. Therefore, should the central server continuously draw the clients whose data is "atypical", the selection bias would result in a model which has been trained on many samples which are more likely to be memorised. As a result, in decentralised settings, there are additional considerations such as client sampling and update weighting strategies, which should be taken into account during FL model training.

In addition, as discussed in Thakkar et al. (2020), model aggregation can have a profound effect on memorisation capacity of the global model. In particular, one of the most frequently used aggregation strategies, namely the federated averaging, was shown to reduce the unintended memorisation (measured using canary-based techniques proposed in Carlini et al. (2019b)). Additionally, the choice of aggregation can have a direct implications on the privacy of the individuals in a decentralised setting (as shown in Geiping et al. (2020); Gupta et al. (2022)), meaning that memorisation of sensitive data can have a much higher impact on the safety of the training data. These results open up a promising area of future research in the area of decentralised memorisation and its interplay with privacy: "Are certain aggregation strategies preferable for guiding the memorisation process on both the local and the global levels?" While there is some evidence showing that even the simplest aggregation algorithms can have an effect, more concrete research is needed in order to establish clear guidelines on this matter.

