# OpenReview forum: "Memorisation in Machine Learning: A Survey of Results"
_TMLR — Accepted by TMLR_

### Review · Reviewer_TV1U · 2024-05-18

**Summary Of Contributions:**

The paper surveys memorisation in machine learning, drawing connections to several related topics, such as, influence functions, data valuation, gradient-related measures, information theory, and privacy. The paper presents a nice and clear overview of the topic and of the related subjects, highlighting connections, some disagreements and conflicting claims in existing literature, as well as some potential directions for future research.

**Audience:**

Yes

**Broader Impact Concerns:**

I have no broader concerns with this paper.

**Claims And Evidence:**

Yes

**Requested Changes:**

1) Can you clarify the use of the term "memorization" as used in different parts of the paper; maybe a table of the possible variants could help?

2) The Key points from Sec 5.2 seem unfounded considering what is actually discussed in the text. Consider either re-wording the Key points or moving this, e.g., under Sec 6, where it has more support.

3) Last part of Sec 5.1, paragraph Grads w.r.t. weights "We note that..."might fit better under the next paragraph Grads w.r.t. inputs?

4) Sec. 9.1, p.19: "Authors of Jagielski et al. (2022) argue that similarly to the conclusions of Feldman (2020) that ML models have to memorise, they also have to forget." Please add some additional explanation or clarification to this: is this a theorem, a conjecture, empirical observation or what?

5) Sec. 9.3, p.20: "While applying DP to state-of-the-art ML models comes with a utility penalty (which may be unavoidable due to the fact that DP limits memorisation)." In some sense we already know that a hit on utility is generally necessary for any privacy protection scheme that aims to avoid catastrophic privacy leakage from the results of Dinur and Nissim (2003, Revealing Information while Preserving Privacy). Please clarify this.

6) Sec. 10: it might be helpful to add cites supporting the recommendation to the list, so it would be easier to check the evidence for these.

7) Sec 5.3, p11: is V-usable information the same as V-information discussed in Sec 5.3? If so, please unify the spelling to make this clear.

Also, please add the following references to the discussion (or comment on why they are not relevant):

Chatterjee 2018: Learning and Memorization.
Jiang et al. 2021: Characterizing Structural Regularities of Labeled Data in Overparameterized Models.
Naseh et al. 2023: Understanding (Un)Intended Memorization in Text-to-Image Generative Models.
Wang et al. 2024: Memorization in self-supervised learning improves downstream generalization.

**Strengths And Weaknesses:**

### Strengths

* The paper is mostly very nice to read and clear on the topic.
* The paper gives a nice overview of the closely related topics, describing how they connect to memorization and what sets them apart.
* While it does not cite all the possible papers in all of the topics covered (which I think is reasonable given the scope of the work), I cannot find many obvious omissions in the cited work.


### Weaknesses

i) The main weakness in the paper is that the term memorization has several different meanings. While the origin of the problem is in the existing literature, which does not currently agree on a single definition, this can make it convoluted at times to keep track of the exact sense in which the term is used in different parts of the manuscript; whether it is self-influence, the ability to memorize some samples exactly (which can be measured, e.g., by using canaries in generative models), or some less-well defined general sense in which, for example, MIAs can be used to quantify how much of the data a given model has memorized.

ii) Besides the above, I mostly have some minor questions and comments (see below in Requested changes).

---

> ### Author Response · Authors · 2024-07-04
> **Response to reviewer TV1U (1)**
>
> >The main weakness in the paper is that the term memorization has several different meanings. While the origin of the problem is in the existing literature, which does not currently agree on a single definition, this can make it convoluted at times to keep track of the exact sense in which the term is used in different parts of the manuscript; whether it is self-influence, the ability to memorize some samples exactly (which can be measured, e.g., by using canaries in generative models), or some less-well defined general sense in which, for example, MIAs can be used to quantify how much of the data a given model has memorized.
>
> We thank the reviewer for raising this point. We note that this exact issue is what our work is trying to address, as there are in fact many different formulations of what memorisation is. Nonetheless, we agree that in this work we can make it more explicit with respect to the definitions we discuss. We address this comment in more detail in our response above.
>
> >Can you clarify the use of the term "memorization" as used in different parts of the paper; maybe a table of the possible variants could help?
>
> We thank the reviewer for this comment. Please note that in this work we reserve the word ‘memorisation’ for a specific meaning it was assigned in formulation proposed by Feldman et al. [1] and the follow-up work by Zhang et al. [2], which is the definition under the prism of self-influence presented in Section 3. We do agree that as the community has been using many different interpretations of this term in prior works, some clarification may be required when distinguishing between the operational definitions of memorisation. However, we argue that while the specific interpretations of this term may be different, the phenomenon that the authors attempt to describe is, in fact, the same (and is based on the change in model’s performance on specific samples when they are included/excluded from the training dataset, i.e. replicating the notion of the self-influence). We have adapted the wording throughout the manuscript sections to make this more clear.
>
> [1] - Feldman, Vitaly, and Chiyuan Zhang. "What neural networks memorize and why: Discovering the long tail via influence estimation." Advances in Neural Information Processing Systems 33 (2020): 2881-2891.
>
> [2] - Zhang, Chiyuan, et al. "Counterfactual memorization in neural language models." Advances in Neural Information Processing Systems 36 (2023): 39321-39362.
>
> >The Key points from Sec 5.2 seem unfounded considering what is actually discussed in the text. Consider either re-wording the Key points or moving this, e.g., under Sec 6, where it has more support.
>
> We thank the reviewer for this remark. We have now reworded the key points and Section 5.2 and 6 now read as follows:
>
> Section 5.2
>
> *Techniques from the domain of information theory can identify the samples which are likely to be memorised by considering the amount of information that these samples contain (and, hence, the information that can be learnt by the model). Many of these approaches can, however, suffer from unrealistic assumptions and poor computational performance.*
>
> Section 6:
>
> *Memorisation is a process that can be localised both spatially and temporally (e.g. at a specific point of training or the layer of the model).*
>
> >Last part of Sec 5.1, paragraph Grads w.r.t. weights "We note that..."might fit better under the next paragraph Grads w.r.t. inputs?
>
> We thank the reviewer for this comment, and have now moved this paragraph as requested.

---

> > ### Author Response · Authors · 2024-07-04
> > **Response to reviewer TV1U (2)**
> >
> > >Sec. 9.1, p.19: "Authors of Jagielski et al. (2022) argue that similarly to the conclusions of Feldman (2020) that ML models have to memorise, they also have to forget." Please add some additional explanation or clarification to this: is this a theorem, a conjecture, empirical observation or what?
> >
> > We thank the reviewer for this question. In the work of Jagielski et al. (Section 5), authors empirically demonstrate that forgetting must occur when non-determinism is present during model training. As deep learning often requires multiple sources of non-determinism to be present (e.g. random batch shuffling, stochastic gradient methods etc.), these are, therefore causing the models to forget. Authors additionally demonstrate that should non-determinism be eliminated in full, forgetting no longer occurs. This is similarly to the conclusions of Feldman et al., who theoretically demonstrate that in order to ensure high model performance, ML models are required to memorise certain data points (and this is, in turn, affected by sources of non-determinism, limiting the amount of information that can be memorised through e.g. differentially private training and the noise it introduces).
> >
> > We have now amended Section 9.1 to include this discussion, which now reads as follows:
> >
> > *Authors of [1] argue that similarly to the conclusions of [2] that ML models have to memorise, they also have to forget.
> > As deep learning often requires multiple sources of non-determinism to be present (e.g. random batch shuffling, stochastic gradient methods etc.), these are, therefore causing the models to forget.
> > Authors additionally demonstrate that should non-determinism be eliminated in full, forgetting no longer occurs, which is ‘’in spirit’ similar to the conclusions of [2].
> > In [2] authors formalise that if a source of non-determinism is included during training (e.g. addition of noise under differential privacy), then the models would be limited in the amount of information they can memorise (and, hence, forget).*
> >
> > [1] - Jagielski, Matthew, et al. "Measuring forgetting of memorized training examples." arXiv preprint arXiv:2207.00099 (2022).
> >
> > [2] - Feldman, Vitaly, and Chiyuan Zhang. "What neural networks memorize and why: Discovering the long tail via influence estimation." Advances in Neural Information Processing Systems 33 (2020): 2881-2891.
> >
> > >Sec. 9.3, p.20: "While applying DP to state-of-the-art ML models comes with a utility penalty (which may be unavoidable due to the fact that DP limits memorisation)." In some sense we already know that a hit on utility is generally necessary for any privacy protection scheme that aims to avoid catastrophic privacy leakage from the results of Dinur and Nissim (2003, Revealing Information while Preserving Privacy). Please clarify this.
> >
> > We thank the reviewer for this remark. We agree that this conclusion has previously been established, and in this specific paragraph we wanted to reiterate that one of the fundamental reasons behind this loss of utility in the context of ML is the reduction in the model's memorisation capacity under DP.
> >
> > We have now reworded Section 9.3 to clarify this further, which reads as follows:
> >
> > *It is known that any privacy protection scheme which does not allow for catastrophic privacy degradation incurs some utility penalty [1]. While applying DP in the context of ML also results in utility penalty (which may be unavoidable due to the fact that DP limits memorisation, as shown in [2,3]), recent works have demonstrated that, for many real-world datasets, competitive performance can still be achieved by applying a number of training adaptations such as pre-training on large public datasets [4,5].*
> >
> > [1] - Dinur, Irit, and Kobbi Nissim. "Revealing information while preserving privacy." Proceedings of the twenty-second ACM SIGMOD-SIGACT-SIGART symposium on Principles of database systems. 2003.
> >
> > [2] - Feldman, Vitaly, and Chiyuan Zhang. "What neural networks memorize and why: Discovering the long tail via influence estimation." Advances in Neural Information Processing Systems 33 (2020): 2881-2891.
> >
> > [3] - Wang, Wenhao, et al. "Memorization in self-supervised learning improves downstream generalization." arXiv preprint arXiv:2401.12233 (2024).
> >
> > [4] - De, Soham, et al. "Unlocking high-accuracy differentially private image classification through scale." arXiv preprint arXiv:2204.13650 (2022).
> >
> > [5] - Berrada, Leonard, et al. "Unlocking accuracy and fairness in differentially private image classification." arXiv preprint arXiv:2308.10888 (2023).

---

> > > ### Author Response · Authors · 2024-07-04
> > > **Response to reviewer TV1U (3)**
> > >
> > > >Sec. 10: it might be helpful to add cites supporting the recommendation to the list, so it would be easier to check the evidence for these.
> > >
> > > We thank the reviewer for this suggestion. We have now added the relevant citations to Section 10.
> > >
> > >
> > > >Sec 5.3, p11: is V-usable information the same as V-information discussed in Sec 5.3? If so, please unify the spelling to make this clear.
> > >
> > > We thank the reviewer for this question. These notions are, in fact, identical (and as per Ethayarajh et al., [1] ‘used interchangeably’). We have now clarified this in Section 5.3, which reads as follows:
> > >
> > > *This motivated the development of usable information theory and its variant of information termed V-information (or V-usable information) [2].*
> > >
> > > [1] - Ethayarajh, Kawin, Yejin Choi, and Swabha Swayamdipta. "Understanding Dataset Difficulty with $\mathcal {V} $-Usable Information." International Conference on Machine Learning. PMLR, 2022.
> > >
> > > [2] - Xu, Yilun, et al. "A theory of usable information under computational constraints." arXiv preprint arXiv:2002.10689 (2020).
> > >
> > > >Also, please add the following references to the discussion (or comment on why they are not relevant):
> > > Chatterjee 2018: Learning and Memorization. Jiang et al. 2021: Characterizing Structural Regularities of Labeled Data in Overparameterized Models. Naseh et al. 2023: Understanding (Un)Intended Memorization in Text-to-Image Generative Models. Wang et al. 2024: Memorization in self-supervised learning improves downstream generalization.
> > >
> > > We thank the reviewer for this suggestion. These works are, indeed, relevant to our discussion! We have added these to relevant Sections: Chatterjee to Section 1, Jiang et al.  to Section 4.1, Naseh et al. to Section 8.2 and Wang et al. to Section 9.3.
> > >
> > > These Sections now read as follows:
> > >
> > > Section 1
> > >
> > > *To the contrary, it appears that both, memorisation and generalisation, are crucial to the learning process of ML models. Prior works lend credence to this fact. An early work by Chatterjee demonstrates that a network of support-limited lookup tables can memorise enough patterns to be able to generalise to those it has never previously encountered.*
> > >
> > > Section 4.1
> > >
> > > *A (somewhat) similar approach was proposed by Jiang et al., termed the per-sample generalisation, measured using a consistency score, which estimates the expected accuracy of the model on a single sample, by comparing the performances of models trained on sets of varying sizes sampled from the data distribution.
> > > This approach, however, also suffers from computational complexity and a number of proxy metrics, such as the pairwise sample distance were proposed to approximate this value.*
> > >
> > > Section 8.2:
> > >
> > > *This question becomes even more challenging to answer, as the community often distinguishes between the ‘intended’ (i.e. the one that contributes towards generation of useful facts) and ‘unintended’ (i.e. the one that reveals PII during generation) memorisation, making it difficult to identify which type of memorisation is a given definition capturing [1,2].*
> > >
> > > [1] - Carlini, Nicholas, et al. "Extracting training data from large language models." 30th USENIX Security Symposium (USENIX Security 21). 2021.
> > >
> > > [2] - Naseh, Ali, Jaechul Roh, and Amir Houmansadr. "Understanding (Un) Intended Memorization in Text-to-Image Generative Models." arXiv preprint arXiv:2312.07550 (2023).
> > >
> > > Section 9.3
> > >
> > > *While applying DP to state-of-the-art ML models comes with a utility penalty (which may be unavoidable due to the fact that DP limits memorisation, as shown in [1,2]), recent works have demonstrated that, for many real-world datasets, competitive performance can be achieved …*
> > >
> > > [1] -  Feldman, Vitaly, and Chiyuan Zhang. "What neural networks memorize and why: Discovering the long tail via influence estimation." Advances in Neural Information Processing Systems 33 (2020): 2881-2891.
> > >
> > > [2] - Wang, Wenhao, et al. "Memorization in self-supervised learning improves downstream generalization." arXiv preprint arXiv:2401.12233 (2024).

---

> ### Comment · Reviewer_TV1U · 2024-07-15
> **Please post a fixed version pdf**
>
> Thanks for the clarifications. It would be helpful if you can submit a corrected draft with the changes highlighted to make re-checking this faster.

---

> > ### Author Response · Authors · 2024-07-15
> > **Request for the highlighted changes**
> >
> > We thank the reviewer for this remark, the file has now been uploaded under the supplementary materials with all changes highlighted.

---

> > > ### Comment · Reviewer_TV1U · 2024-07-18
> > > **Some further comments**
> > >
> > > Thank you for the update, here are some additional comments, in decreasing order of importance:
> > >
> > > * abstract: "we unify a broad range of previous definitions and perspectives on memorisation in ML". While there is plenty of discussion on the existing definitions, and you argue for the self-influence definition, I do not see how this would unify the definitions. Compare this claim to one of the future directions (p24, Standardisation of the notion), which gives the impression that such unification is still to be done.
> > >
> > > * p21: "Authors of Jagielski et al. (2022) argue that similarly to the conclusions of Feldman (2020) that ML models have to memorise, they also have to forget..." I am not sure I still understand this paragraph, is the idea that in order to generalize well, models have to memorize and also forget?
> > >
> > > * p21 Key points: why do you claim that duplicate data points are in particular danger of being forgotten if they are not encountered again? Does not having duplicates by definition mean that the sample is encountered several times?
> > >
> > > * p22: typo "This is also directly related to the concept of model’s stability described in , which is immediate"
> > >
> > > * There are plenty of cite vs citep vs citet errors, please fix ti improve readability.

---

> > > > ### Author Response · Authors · 2024-07-20
> > > > **Response to further comments from reviewer TV1U**
> > > >
> > > > We thank the reviewer for the additional comment and provide a comment-by-comment response below.
> > > >
> > > > >abstract: "we unify a broad range of previous definitions and perspectives on memorisation in ML". While there is plenty of discussion on the existing definitions, and you argue for the self-influence definition, I do not see how this would unify the definitions. Compare this claim to one of the future directions (p24, Standardisation of the notion), which gives the impression that such unification is still to be done.
> > > >
> > > > We thank the reviewer for this comment. We agree that the word ‘unify’ may seem misleading in this scenario. We have changed the wording and indeed, unification is still an open question. In this manuscript we focus on a set of definitions focused around self-influence, which is flexible and has the potential to become a basis for such unification, in particular since there is evidence that some of the works which interpret memorisation as leakage find that higher memorisation in the sense of Feldman also leads to higher adversarial vulnerability [1].
> > > >
> > > > [1] - Choi, Jihye, et al. "Why train more? effective and efficient membership inference via memorization." arXiv preprint arXiv:2310.08015 (2023).
> > > >
> > > > We have now updated the Abstract which reads as follows:
> > > >
> > > > *Quantifying the impact of individual data samples on machine learning models is an open research problem. This is particularly relevant when complex and high-dimensional relationships have to be learned from a limited sample of the data generating distribution, such as in deep learning. It was previously shown that, in these cases, models rely not only on extracting patterns which are helpful for generalisation, but also seem to be required to incorporate some of the training data more or less as is, in a process often termed memorisation. This raises the question: if some memorisation is a requirement for effective learning, what are its privacy implications? In this work we consider a broad range of previous definitions and perspectives on memorisation in ML, discuss their interplay with model generalisation and their implications of these phenomena on data privacy. We then propose a framework to reason over what memorisation means in the context of ML training under the prism of individual sample's influence on the model. Moreover, we systematise methods allowing practitioners to detect the occurrence of memorisation or quantify it and contextualise our findings in a broad range of ML learning settings. Finally, we discuss memorisation in the context of privacy attacks, differential privacy and adversarial actors.*
> > > >
> > > > >p21: "Authors of Jagielski et al. (2022) argue that similarly to the conclusions of Feldman (2020) that ML models have to memorise, they also have to forget..." I am not sure I still understand this paragraph, is the idea that in order to generalize well, models have to memorize and also forget?
> > > >
> > > > We thank the reviewer for this question. This statement was supposed to highlight that there is a similarity between memorisation and forgetting in the sense that both are phenomena that inevitably occur during ML training. To paint a more concrete picture: in order to generalise well, the models have to memorise. And while the model is being trained, it will experience forgetting as well as memorisation at certain training stages.
> > > >
> > > > We have now reworded the paragraph to make this clearer and it reads as follows:
> > > >
> > > > *It was previously shown by [1] that models have to memorise certain information about individual samples over the course of model training.
> > > > Authors of [2] went further and argued that similarly to how models have to memorise, they also have to forget information about individual samples if they are not encountered again during later stages of training.*
> > > >
> > > > [1] - Feldman, Vitaly. "Does learning require memorization? a short tale about a long tail." Proceedings of the 52nd Annual ACM SIGACT Symposium on Theory of Computing. 2020.
> > > >
> > > > [2] - Jagielski, Matthew, et al "Measuring forgetting of memorized training examples." arXiv preprint arXiv:2207.00099 (2022).
> > > >
> > > > >p21 Key points: why do you claim that duplicate data points are in particular danger of being forgotten if they are not encountered again? Does not having duplicates by definition mean that the sample is encountered several times?
> > > >
> > > > We thank the reviewer for raising this point. This was indeed overlooked, the duplicate and atypical samples are both more prone to memorisation, but only the atypical samples are also prone to being forgotten. We apologise for the oversight. This has been corrected.
> > > >
> > > > >p22: typo "This is also directly related to the concept of model’s stability described in , which is immediate"
> > > >
> > > > We thank the reviewer for pointing this out and have fixed the missing citation.
> > > >
> > > > >There are plenty of cite vs citep vs citet errors, please fix it to improve readability.
> > > >
> > > > We thank the reviewer for this comment. We have gone over the manuscript again and corrected these.

---

### Review · Reviewer_HSGv · 2024-05-28

**Summary Of Contributions:**

The paper provides a summary of literature on memorization in machine learning, where models incorporate training data "as is" rather than extracting generalisable patterns. This raises data privacy concerns, as memorized data can be identified or reconstructed. The authors unify definitions and perspectives on memorisation, discuss its relationship with model generalization, and provide methods for detecting and quantifying memorisation. They also explore implications for privacy attacks and differential privacy, and give recommendations to practitioners for mitigating risks.

**Audience:**

Yes

**Broader Impact Concerns:**

None.

**Claims And Evidence:**

Yes

**Requested Changes:**

1. It would be great to give an in-depth overview of the definitions of memorization, and their connections and then argue why one is more suitable than the other. I find the discussion on fitting random labels limited. Say, for instance, one only cares about supervised learning, in that case, is the random labels based definition most appropriate?


2. Clarify to whom is the memorization attributed to (who is said to memorize), model $f$ or algorithm $\mathcal{A}$ or to both? Is there any result (bound) connecting model capacity and memorization?


3. Add discussion on stability ( Bousquet et al. 2002) and its connections with the main definition of memorization.

**Strengths And Weaknesses:**

### Strengths

1. The paper provides a thorough and systematic review of the concept of memorization in machine learning, covering its definitions, their implications, and techniques for detection and prevention. The main focus is on the self and cross influence based definitions that are task and domain agnostic.


2. In addition to the influence-based definitions and their connection to long-tail learning, the paper covers techniques for measuring memorization through influence estimation, its connections to data valuation, and quantities related to memorization such as gradient-based influence proxies and information theoretic quantities. The paper also discusses localization and timing of memorization, inducing deliberate memorization and relation to adversarial samples, and differential privacy.


3. Memorization is a tricky topic with its pros and cons in learning a better model and privacy. The paper covers all of these aspects nicely, rather than favoring a purely negative or positive side of it.  Based on their literature review the authors provide valuable guidelines for practitioners.




### Weaknesses

1. The breadth of topics covered is appreciated. The paper does not go deeper into the definitions.


2. The paper favors the influence-based definition and it appears as if this definition is being posed as the standard definition moving forward. This definition is quantifying the importance of the sample in learning the desired function. But importance does not imply that the sample should be “memorized” and there are several other measures of sample importance (for instance in active learning).  To me, the influence-based definition does not appear to be a satisfactory definition of memorization. Moreover, in eq (3), it is not clear if the memorization is defined for the algorithm, for the model or for both. This is also closely related to the notion of stability ( Bousquet et al. 2002), which should be discussed here.

---

> ### Author Response · Authors · 2024-07-04
> **Response to reviewer HSGv (1)**
>
> > It would be great to give an in-depth overview of the definitions of memorization, and their connections and then argue why one is more suitable than the other. I find the discussion on fitting random labels limited. Say, for instance, one only cares about supervised learning, in that case, is the random labels based definition most appropriate?
>
>
> We thank the reviewer for this remark. We address our choice of definition of memorisation above.
> In essence, we note that it is not Feldman's exact method (i.e. the memorisation through self-influence) per se, but the underlying theory that we regard as the correct way to measure memorisation in the current ML landscape. We further highlight that a lot of the other metrics, which are used to quantify memorisation, actually measure some of the properties of samples which sometimes allegedly cause memorisation, rather than memorisation itself (e.g. sample difficulty, usable information etc.).
> This being said, we appreciate that this might not be immediately clear from the way we formulate our approach in the manuscript. We have now taken reviewers’ suggestions and expanded Section 3 to further highlight our reasoning.
> The Section now reads as follows:
>
> *For a long time, memorisation lacked a precise definition and the term was commonly used loosely to refer to a variety of phenomena.
> In this section, we discuss the definition of [1], who presented the first unified formulation and theory of memorisation in ML.
> We note that in this work we primarily concentrate on the underlying intuition behind the definition of [1], rather than its specific operationalisation.
> While various other definitions have previously been proposed (and we discuss them in great detail in the following sections), we identify that the one proposed in [1], which is based on the notion of influence as the only one which quantifies the phenomenon of memorisation itself, rather than measuring some related phenomena often associated with samples which can be memorised.
> We additionally note that due to the choice of the influence-based formulation, this definition is also a) modality- b) training setting- and c) model-agnostic, making it an attractive generic formulation of memorisation.
> This is primarily because the notion of influence can be easily extended to any learning scenario allowing the user to select which metric is used to quantify how the presence or absence of a training point affects the resulting utility (e.g. per-sample accuracy in classification or log-perplexity in language modelling).
> In contrast, most of the previously proposed definitions which we discuss in this manuscript are either data- or model-specific (e.g. quantifying memorisation via canaries); only apply to specific training settings (e.g. fitting of random labels) or capture a different, albeit related phenomenon (e.g. memorisation through overfitting).
> Fundamentally, this definition is also very intuitive: if humans memorise a sample, they are expected to be able to make a more accurate prediction on it compared to a setting where they have never seen this exact sample before and only extrapolate from our knowledge of similar-looking data.
> Such memorisation may even be required when the data looks so out-of-the-ordinary that there is no straightforward way to break it down into previously seen simple patterns that can be placed into the existing knowledge base.
> Methods which do not directly measure memorisation (but, instead, certain factors leading to it) all leverage this fact: the samples which fall under this behaviour are distinct and individual (or atypical).
> While this was previously implicitly mentioned in prior works on memorisation, only in [1] it was made explicit that memorisation is a phenomenon whereby the learning process of a model is benefitted by contributions/information contained in individual samples.
> This unifies the previous work on these metrics (which we later describe as methods that measure factors leading to memorisation) under this common thread.
> Our claim is that [1] is the first work to formally characterise this phenomenon and offer a method to express it quantitatively and irrespective of the learning setting.*
>
> [1] - Feldman, Vitaly. "Does learning require memorization? a short tale about a long tail." Proceedings of the 52nd Annual ACM SIGACT Symposium on Theory of Computing. 2020.

---

> > ### Author Response · Authors · 2024-07-04
> > **Response to reviewer HSGv (2)**
> >
> > > Clarify to whom is the memorization attributed to (who is said to memorize), model $f$ or algorithm $\mathcal{A}$ or to both? Is there any result (bound) connecting model capacity and memorization?
> >
> > We would like to thank the reviewer for this comment. The algorithm here represents the training process of the model, during which memorisation occurs, but it is ultimately the model itself that memorises. We have now amended the Section, which reads as follows:
> >
> > *The algorithm here represents the training process of the model, during which memorisation occurs, but it is ultimately the model itself that memorises. Note that a naive calculation of Eq. 3 is computationally expensive*.
> >
> > To answer the second question: there exists a number of works making the connection between the two concepts, but the final conclusions are not yet established. Partially this is due to the fact that model capacity has (similarly to memorisation) multiple interpretations. The one we refer to in this work is the model's capacity for memorisation (in Fledman’s definition), but other definitions exist (e.g. Zhang et al.’s work on fitting random labels).
> >
> > For instance, the authors of [1] argue that larger (i.e. in this definition more overparametrised) models have higher memorisation capacity, leading to larger models being more vulnerable to adversarial influence (and data extraction attacks in particular). Similarly, authors of [2] show that the model's ability to memorise grows with larger capacity. Finally, authors of [3] also claim that memorisation grows as the capacity of the model is increased. Therefore while there certainly are connections between the two (and in the majority of works is in favour of the argument that higher model capacity is associated with higher degree of memorisation), these are not yet causal (i.e. there is no established formal connection between higher capacity necessarily leading to higher memorisation). We suspect that this is partially the case due to the fact that what is meant by memorisation in these settings differs.
> >
> > [1] - Carlini, Nicholas, et al. "The secret sharer: Evaluating and testing unintended memorization in neural networks." 28th USENIX security symposium (USENIX security 19). 2019.
> >
> > [2] - Meehan, Casey, et al. "Do SSL Models Have Déjà Vu? A Case of Unintended Memorization in Self-supervised Learning." Advances in Neural Information Processing Systems 36 (2024).
> >
> > [3] - Carlini, Nicholas, et al. "Quantifying memorization across neural language models." arXiv preprint arXiv:2202.07646 (2022).
> >
> > We have expanded Section 8.2 to include this discussion, which now reads as follows:
> >
> >
> > *Moreover, given the recent advances in the field of LLMs with many models exceeding billions of trainable parameters[1,2], it was shown that as the model's size (and its effective capacity) grows, so does the capacity for memorisation [3].
> > While no formal relationship between model's effective capacity and its capacity to memorise has been established, most prior evidence shows that these are, in fact, positively correlated, presenting additional privacy challenges when training large generative models on sensitive data [3,4,5].*
> >
> > [1] - Touvron, Hugo, et al. "Llama 2: Open foundation and fine-tuned chat models." arXiv preprint arXiv:2307.09288 (2023).
> >
> > [2] - Jiang, Albert Q., et al. "Mixtral of experts." arXiv preprint arXiv:2401.04088 (2024).
> >
> > [3] - Carlini, Nicholas, et al. "The secret sharer: Evaluating and testing unintended memorization in neural networks." 28th USENIX security symposium (USENIX security 19). 2019.
> >
> > [4] - Meehan, Casey, et al. "Do SSL Models Have Déjà Vu? A Case of Unintended Memorization in Self-supervised Learning." Advances in Neural Information Processing Systems 36 (2024).
> >
> > [5] - Carlini, Nicholas, et al. "Quantifying memorization across neural language models." arXiv preprint arXiv:2202.07646 (2022).

---

> > > ### Author Response · Authors · 2024-07-04
> > > **Response to reviewer HSGv (3)**
> > >
> > > > Add discussion on stability ( Bousquet et al. 2002) and its connections with the main definition of memorization.
> > >
> > > We thank the reviewer for this suggestion and agree that there is, in fact, a clear connection between the two notions. The definition of stability discussed in Bousquet et al. is directly related to the definition of sensitivity used in literature on DP as well as Feldman’s discussion on memorisation. DP is capable of bounding the amount of information the model can learn from a given sample, this is typically achieved by applying randomised noise to the output of the query (in our case training algorithm). In the context of ML, this is typically achieved through DP-SGD, where the noise is added to the gradient of the model. The amount of noise that needs to be added is determined by the sensitivity of a function (in the context of DP-SGD it can be obtained through gradient clipping, bounding the sensitivity of an individual sample). Therefore, by bounding the sensitivity of individual samples, one can bound their leave-one-out stability and, hence, the algorithms that memorise less under DP are considered to be stable in Feldman.
> > >
> > > We have now expanded Section 9.3 to include this. The Section now reads as follows:
> > >
> > > *This is also directly related to the concept of model's stability described in [1], which is immediate from the results of [2] and [3]. In particular, the former studies the relationship between the leave-one-out stability of an individual sample and model's memorisation capacity, showing that under DP, stability increases and memorisation decreases. Thus, informally, stable algorithms memorise less and DP algorithms are ’very stable’, leading to reduced memorisation.*
> > >
> > > [1] - Bousquet, Olivier, and André Elisseeff. "Stability and generalization." The Journal of Machine Learning Research 2 (2002): 499-526.
> > >
> > > [2] - Feldman, Vitaly. "Does learning require memorization? a short tale about a long tail." Proceedings of the 52nd Annual ACM SIGACT Symposium on Theory of Computing. 2020.
> > >
> > > [3] - Brown, Gavin, et al. "When is memorization of irrelevant training data necessary for high-accuracy learning?." Proceedings of the 53rd annual ACM SIGACT symposium on theory of computing. 2021.

---

> > > > ### Comment · Reviewer_HSGv · 2024-07-21
> > > >
> > > > Thanks for the response. I have read it and submitted my recommendation.

---

### Review · Reviewer_PFHo · 2024-06-21

**Summary Of Contributions:**

This paper reviews a number of approaches for quantifying the importance of individual data samples in deep learning. They survey a number of definitions, focusing in particular on those inspired by influence functions, and attempt to formally unify them. They also discuss a bit around questions of if memorization can be localized, and finally discuss privacy and safety implications, as well as providing some recommendations for practitioners.

**Audience:**

Yes

**Claims And Evidence:**

No

**Requested Changes:**

The most two important things would be 1) clarification around the attempt to unify - either making this unification clearer, or adjusting the paper's claims; and 2) ensuring that the "key points" in each section are actually backed up by the text: clearer signposting of the arguments with these points would be helpful on reading.

**Strengths And Weaknesses:**

TL;DR: I learned a lot about a range of work reading this paper, which is a main goal of a review. However, I'm not sure that the attempted synthesis really landed for me.

Main Critiques:
- The paper claims to "unify a broad range of definitions" - I'm not sure that I agree that it successfully does that. Sections 2/3 seem to be the spots where such a synthesis might occur, leaning on a discussion of influence - however, the takeaways are only ever really specified in a way that makes sense for a supervised L, and Section 3 is really very supervised-learning specific. It's possible that some generative models which output likelihoods could fit into this framework, but the rest of the paper makes it clear that sampling-focused models are very important here too. The unsupervised aspect of this is not so important per se, but the headline claim of the paper is this unification and to me I don't totally think it happens. Even in Section 11, the authors list unifying definitions as a highly important future direction - suggesting that contrary to the claim in the abstract, this paper did not perform such unification
- often I found the "key points" to not really reflect the text above them. For instance, at the end of Sec 4.2, this subtle point around whether game theoretic metrics are or are not a proxy for memorization does not seem to be discussed in detail in the text. Other examples: Sec 5.1 - the “key point” states quite strongly that gradient-based metrics are not helpful for quantifying memorization - I think this argument should be made more clearly in the text as it seems important; Sec 5.2: the “key point” around localization is not argued so clearly in the text; Sec 9.1 - the key point is not how I would summarize this section


Secondary Notes:
- Fig 1b is a little confusing to me - I'm not sure it's very helpful in communicating its point. Specifically, there's a "long tail" aspect which I think is supposed to connect to this visual which doesn't really
- In Sec 3.1, you draw a distinction between low-quality/mislabelled samples and "true" long tail samples. It's not clear to me from a pure statistical perspective that this is a real distinction (mislabelled examples that trigger this metric are kind of long tail by definition), and it would be good to get more understanding here of what this really means. This clarification would also help the analogy in Sec 7 between mislabelled examples and canaries)
- in Sec. 3.2, you give two definitions of measuring memorization: Eq 5 which is for a model, and "fitting random labels" which is for a learning algorithm. It seems important to distinguish these
- in Sec 4.2 - it's not clear what LOO influence is (is this an influence function thing or a data shapley thing?)


Smaller points:
- sampling format throughout is quite inconsistent - make sure you're using \citet and \citep appropriately, as it greatly improves readability

- Sec 5: I think a more precise title would be helpful here: "Quantities ostensibly related to memorisation" could describe the entire paper!
- Sec 5.1 - it would be helpful to draw a clearer distinction between IF techniques and techniques around 2nd order derivatives - they're conceptually quite related
- Sec 5.2: I don't think it's ever stated how V-estimation gets around the issue of non-random weights
- Sec 6: in the top paragraph, the authors state: "different parts of the model become responsible ... for learning individual concepts ... thus it is possible to identify individual neurons associated with specific concepts". This argument doesn't quite hold to me - in fact identifying these neurons is quite hard, if they exist at all! This is a huge challenge in the field of mechanistic interpretability.
- p21: "uttermost" -> I think "utmost" is the more common word here
- 7.2 - I would draw a stronger distinction between adversarial examples and data poisoning (which you don't actually name but seems to be some of what you're describing)
- "since pixels (on their own) are independent" - I don't understand this claim, I wouldn't say that pixels are independent in a natural image

---

> ### Author Response · Authors · 2024-07-04
> **Response to reviewer PFHo (1)**
>
> >The paper claims to "unify a broad range of definitions" - I'm not sure that I agree that it successfully does that. Sections 2/3 seem to be the spots where such a synthesis might occur, leaning on a discussion of influence - however, the takeaways are only ever really specified in a way that makes sense for a supervised L, and Section 3 is really very supervised-learning specific. It's possible that some generative models which output likelihoods could fit into this framework, but the rest of the paper makes it clear that sampling-focused models are very important here too. The unsupervised aspect of this is not so important per se, but the headline claim of the paper is this unification and to me I don't totally think it happens. Even in Section 11, the authors list unifying definitions as a highly important future direction - suggesting that contrary to the claim in the abstract, this paper did not perform such unification
>
> We thank the reviewer for this comment. We agree that we take a strong position and highlight the formulation proposed by Feldman as the ‘main’ conceptual definition of memorisation, instead of joining many existing definitions together. This being said, we note that it is not the specific definition we concentrate on, but rather the intuition behind it. Feldman's work also points to a broader, conceptual understanding that is (informally): memorisation is a phenomenon whereby the specific information from individual samples (with an emphasis on individual) rather than the general information that can be learned from any number of generic samples is influential/meaningful to a model's training. This intuitively explains the choice of terminology (humans have to memorise "special" facts which don't fit into "general knowledge"), the connection to individual influence and to notions like DP and leave-one-out generalisation (these restrict/quantify the reliance of models on individual samples). Methods which do not directly measure memorisation (but the things leading to it) all seem to leverage this fact: the samples which fall under this behaviour are distinct and individual, (or "special"). This unifies previous work on these metrics (which we say ostensibly measure memorisation) under this common thread.
> This makes Feldman a very attractive formulation to describe the phenomenon of memorisation.
>
> We do, however, understand that this might not be obvious from the way we currently formulate Section 3 and we wanted to convey that it is the principle discussed in Feldman, rather than the specific interpretation that we want to promote as the means to reason over memorisation.
>
> We have now amended the wording, which reads as follows (next response):

---

> > ### Author Response · Authors · 2024-07-04
> > **Response to reviewer PFHo (2)**
> >
> > *‘For a long time, memorisation lacked a precise definition and the term was commonly used loosely to refer to a variety of phenomena.
> > In this section, we discuss the definition of [1], who presented the first unified formulation and theory of memorisation in ML.
> > We note that in this work we primarily concentrate on the underlying intuition behind the definition of [1], rather than its specific operationalisation.
> > While various other definitions have previously been proposed (and we discuss them in great detail in the following sections), we identify that the one proposed in [1], which is based on the notion of influence as the only one which quantifies the phenomenon of memorisation itself, rather than measuring some related phenomena often associated with samples which can be memorised.
> > We additionally note that due to the choice of the influence-based formulation, this definition is also a) modality- b) training setting- and c) model-agnostic, making it an attractive generic formulation of memorisation.
> > This is primarily because the notion of influence can be easily extended to any learning scenario allowing the user to select which metric is used to quantify how the presence or absence of a training point affects the resulting utility (e.g. per-sample accuracy in classification or log-perplexity in language modelling).
> > In contrast, most of the previously proposed definitions which we discuss in this manuscript are either data- or model-specific (e.g. quantifying memorisation via canaries); only apply to specific training settings (e.g. fitting of random labels) or capture a different, albeit related phenomenon (e.g. memorisation through overfitting).
> > Fundamentally, this definition is also very intuitive: if humans memorise a sample, they are expected to be able to make a more accurate prediction on it compared to a setting where they have never seen this exact sample before and only extrapolate from our knowledge of similar-looking data.
> > Such memorisation may even be required when the data looks so out-of-the-ordinary that there is no straightforward way to break it down into previously seen simple patterns that can be placed into the existing knowledge base.
> > Methods which do not directly measure memorisation (but, instead, certain factors leading to it) all leverage this fact: the samples which fall under this behaviour are distinct and individual (or atypical).
> > While this was previously implicitly mentioned in prior works on memorisation, only in [1] it was made explicit that memorisation is a phenomenon whereby the learning process of a model is benefitted by contributions/information contained in individual samples.
> > This unifies the previous work on these metrics (which we later describe as methods that measure factors leading to memorisation) under this common thread.
> > Our claim is that [1] is the first work to formally characterise this phenomenon and offer a method to express it quantitatively and irrespective of the learning setting.’*
> >
> > [1] - Feldman, Vitaly. "Does learning require memorization? a short tale about a long tail." Proceedings of the 52nd Annual ACM SIGACT Symposium on Theory of Computing. 2020.

---

> > > ### Author Response · Authors · 2024-07-04
> > > **Response to reviewer PFHo (3)**
> > >
> > > >often I found the "key points" to not really reflect the text above them. For instance, at the end of Sec 4.2, this subtle point around whether game theoretic metrics are or are not a proxy for memorization does not seem to be discussed in detail in the text. Other examples: Sec 5.1 - the “key point” states quite strongly that gradient-based metrics are not helpful for quantifying memorization - I think this argument should be made more clearly in the text as it seems important; Sec 5.2: the “key point” around localization is not argued so clearly in the text; Sec 9.1 - the key point is not how I would summarize this section
> > >
> > > We thank the reviewer for these remarks. We have now substantially changed the key points (and in some cases the Sections they are meant to summarise), which read as follows:
> > >
> > > Section 4.2
> > >
> > > Main text:
> > >
> > > *In many cases, the data samples which are ‘valued higher’ are often the same ones that are more likely to be memorised [1].
> > > …
> > > As a result, while there are some clear overlaps between the domains of memorisation and data valuation, there are still many open questions (particularly on the relationship of the valuation metrics, which often concentrate on the cross-influence and the self-influence discussed in the context of memorisation) making the use of data valuation techniques for quantification of memorisation problematic.*
> > >
> > > [1] - Zhang, Hao, et al. "Interpreting multivariate shapley interactions in dnns." Proceedings of the AAAI Conference on Artificial Intelligence. Vol. 35. No. 12. 2021.
> > >
> > > Key point:
> > >
> > > *Game-theoretic data valuation metrics assign higher values to samples which are often more likely to be memorised, but are not a direct proxy for memorisation.*
> > >
> > > Section 5.1
> > >
> > > Main text:
> > >
> > > *Overall, many properties of training data which can be efficiently represented through the gradients (e.g. the privacy loss) are often correlated to the magnitude of memorisation of that training sample.
> > > Therefore many of the samples that possess e.g. higher gradient norms are often those that are also more likely to be memorised.
> > > However, these links are mostly correlational as to-date there has not been any empirical or theoretical evidence to show that there is a causal relationship between the characteristics of the gradient and the memorisation of individual samples.
> > > Hence, we argue that the gradient-derived metrics cannot be used as direct approximations of memorisation.*
> > >
> > > Key point:
> > >
> > > *Gradient-based metrics cannot be used to directly quantify memorisation, but often (efficiently) measure related quantities (e.g. sample difficulty).*
> > >
> > > Section 5.2
> > >
> > > *Techniques from the domain of information theory can identify the samples which are likely to be memorised by considering the amount of information that these samples contain (and, hence, the information that can be learnt by the model). Many of these approaches can, however, suffer from unrealistic assumptions and poor computational performance.*
> > >
> > > Section 9.1
> > >
> > > *Samples that are memorised during training can be forgotten if they are not encountered again. This is particularly common for atypical or duplicate data points.*
> > >
> > > >Fig 1b is a little confusing to me - I'm not sure it's very helpful in communicating its point. Specifically, there's a "long tail" aspect which I think is supposed to connect to this visual which doesn't really
> > >
> > > We thank the reviewer for this comment. We would be grateful if the reviewer could specify what they mean and we would be happy to revise the figure.

---

> > > > ### Author Response · Authors · 2024-07-04
> > > > **Response to reviewer PFHo (4)**
> > > >
> > > > >In Sec 3.1, you draw a distinction between low-quality/mislabelled samples and "true" long tail samples. It's not clear to me from a pure statistical perspective that this is a real distinction (mislabelled examples that trigger this metric are kind of long tail by definition), and it would be good to get more understanding here of what this really means. This clarification would also help the analogy in Sec 7 between mislabelled examples and canaries)
> > > >
> > > > We thank the reviewer for this remark. We would like to highlight that this distinction depends on the data-generating distribution. If the distribution is long-tailed, then a large proportion of the examples sampled from it would look atypical. However, should the distribution not be long-tailed, a mislabelled sample would ‘look to the model’ (and cause the corresponding behaviour) as if it came from the long-tailed distribution as well, but statistically speaking the distributions are different.
> > > >
> > > > To further highlight the difference between the potentially mislabelled long-tailed samples and the rare correctly-labelled samples we consider the difference these have with respect to their impact on the rest of the training population. One theory is that samples with rare features have high positive self-influence and (likely) positive cross-influence. Contrary, malformed samples have high positive self-influence and high negative (i.e. they are able to drastically reduce performance) cross-influence.
> > > >
> > > > To clarify this distinction further we have now expanded the Conclusion, which reads as follows:
> > > >
> > > > *Samples with higher information content tend to be more prone to memorisation. However, as we established, not all information contained in those samples is useful to the model. Moreover, samples which are described as ‘more difficult’ can be either highly informative (but rare) or of poor utility because they are malformed. While both of these sample types have high positive self-influence (i.e. if included, their utility on themselves improves), the former tend to be of high positive cross-influence, (i.e. improving the utility on other samples) and the latter are often of high negative cross-influence, harming the overall performance on other data points. We believe that further work is required to be able to establish strong links between information content, memorisation and the resulting model utility.*
> > > >
> > > > >In Sec 3.1, you draw a distinction between low-quality/mislabelled samples and "true" long tail samples. It's not clear to me from a pure statistical perspective that this is a real distinction (mislabelled examples that trigger this metric are kind of long tail by definition), and it would be good to get more understanding here of what this really means. This clarification would also help the analogy in Sec 7 between mislabelled examples and canaries)
> > > >
> > > > We thank the reviewer for this remark. We would like to highlight that this distinction depends on the data-generating distribution. If the distribution is long-tailed, then a large proportion of the examples sampled from it would look atypical. However, should the distribution not be long-tailed, a mislabelled sample would ‘look to the model’ (and cause the corresponding behaviour) as if it came from the long-tailed distribution as well, but statistically speaking the distributions are different.
> > > >
> > > > To further highlight the difference between the potentially mislabelled long-tailed samples and the rare correctly-labelled samples we consider the difference these have with respect to their impact on the rest of the training population. One theory is that samples with rare features have high positive self-influence and (likely) positive cross-influence. Contrary, malformed samples have high positive self-influence and high negative (i.e. they are able to drastically reduce performance) cross-influence.
> > > >
> > > > To clarify this distinction further we have now expanded the Conclusion, which reads as follows:
> > > >
> > > > *Samples with higher information content tend to be more prone to memorisation. However, as we established, not all information contained in those samples is useful to the model. Moreover, samples which are described as ‘more difficult’ can be either highly informative (but rare) or of poor utility because they are malformed. While both of these sample types have high positive self-influence (i.e. if included, their utility on themselves improves), the former tend to be of high positive cross-influence, (i.e. improving the utility on other samples) and the latter are often of high negative cross-influence, harming the overall performance on other data points. We believe that further work is required to be able to establish strong links between information content, memorisation and the resulting model utility.*

---

> ### Author Response · Authors · 2024-07-04
> **Response to reviewer PFHo (5)**
>
> >in Sec. 3.2, you give two definitions of measuring memorization: Eq 5 which is for a model, and "fitting random labels" which is for a learning algorithm. It seems important to distinguish these
>
> We thank the reviewer for this remark. The algorithm here represents the training process of the model, during which memorisation occurs, but it is ultimately the model itself that memorises. We have now added a comment on this and the Section reads as follows:
>
> *The algorithm here represents the training process of the model, during which memorisation occurs, but it is ultimately the model itself that memorises. Note that a naive calculation of Eq. 3 is computationally expensive*.
>
> >in Sec 4.2 - it's not clear what LOO influence is (is this an influence function thing or a data shapley thing?)
>
> We thank the reviewer for this comment. The concept of the leave-one-out influence was first proposed in [1] and while it is related to the concepts discussed in data Shapley, this is a method proposed specifically for influence estimation.
>
> [1] - Cook, R. D. Detection of influential observation in linear regression. Technometrics, 19(1):15–18, 1977.
>
> >sampling format throughout is quite inconsistent - make sure you're using \citet and \citep appropriately, as it greatly improves readability
> p21: "uttermost" -> I think "utmost" is the more common word here
>
> We thank the reviewer for this remark and have now corrected these.
>
> >Sec 5: I think a more precise title would be helpful here: "Quantities ostensibly related to memorisation" could describe the entire paper!
>
> We thank the reviewer for this suggestion. We have now renamed this section to *‘Metrics to identify factors leading to memorisation’*
>
> >Sec 5.1 - it would be helpful to draw a clearer distinction between IF techniques and techniques around 2nd order derivatives - they're conceptually quite related
>
> We thank the reviewer for this comment. We agree, in principle, that these techniques are similar. It is important to note, however, that influence relies on the weight curvature (i.e. Hessian with respect to model weights), while many other related metrics describe the input space curvature (i.e. Hessian with respect to inputs). Similarly, some methods in the domain of gradient-based functions (Section 5) are related to the input space curvature (e.g. the variance of gradience can be seen as its first-order approximation).
>
> To highlight these subtle points, we have now expanded Section 5.1, which reads as follows:
>
> *While these are conceptually similar to the efficient influence estimations proposed in [1], the subtle difference between the two concerns which type of curvature the authors discuss.
> For influence estimation, the derivatives are taken with respect to the model weights, compared to methods such as [2] (or a first-order approximation proposed in [3]), where the derivatives are taken with respect to the model inputs).*
>
> [1] - Koh, Pang Wei, and Percy Liang. "Understanding black-box predictions via influence functions." International conference on machine learning. PMLR, 2017.
>
> [2] - Garg, Isha, and Kaushik Roy. "Memorization through the lens of curvature of loss function around samples." arXiv preprint arXiv:2307.05831 (2023).
>
> [3] - Agarwal, Chirag, Daniel D'souza, and Sara Hooker. "Estimating example difficulty using variance of gradients." Proceedings of the IEEE/CVF Conference on Computer Vision and Pattern Recognition. 2022.
>
> >Sec 5.2: I don't think it's ever stated how V-estimation gets around the issue of non-random weights
>
> We thank the reviewer for raising this point. The work of [1] on uses the entropy of the softmax (which is a distribution) to calculate the V-information in such settings. We have now added this line to the corresponding Section which reads as follows:
>
> *This method can be seen as a computationally constrained version of Shannon MI, which
> measures the ‘usable’ information contained in the data, which can be extracted by functions (e.g. ML models) in a specific family (denoted V).
> To mitigate the aforementioned issue of non-randomness, [1] proposed to use the entropy of a softmax function (which is in itself a distribution).*
>
> [1] - Xu, Yilun, et al. "A theory of usable information under computational constraints." arXiv preprint arXiv:2002.10689 (2020).

---

> > ### Author Response · Authors · 2024-07-04
> > **Response to reviewer PFHo (6)**
> >
> > >Sec 6: in the top paragraph, the authors state: "different parts of the model become responsible ... for learning individual concepts ... thus it is possible to identify individual neurons associated with specific concepts". This argument doesn't quite hold to me - in fact identifying these neurons is quite hard, if they exist at all! This is a huge challenge in the field of mechanistic interpretability.
> >
> > We thank the reviewer for this remark. We agree with this and we want to highlight that we are not claiming that this is already possible, but rather making a comment on the existing state-of-the-art using the works described in Section 6 as examples and elaborate on the progress made in this direction. We also agree that these results are of direct relevance to the field of mechanistic interpretability, and are happy to include this in our discussion, highlighting that it has previously been used to analyse ML model behaviour (including concepts related to memorisation). To summarise: we are not arguing that this research question has been fully addressed, but rather pointing the reader towards the previous attempts at solving this issue.
> >
> > We have altered the wording in Section 6 to reflect this, which now reads as follows:
> >
> > *Thus, authors argue that it can be possible to identify individual neurons associated with specific concepts.*
> >
> > …
> >
> > *Additionally, [1] previously shown that certain architecture-specific parts of the model may even be ‘responsible’ for memorisation altogether (in this case the attention heads).
> > These findings highlight that learning (and memorisation) is not only inconsistent across different model types (i.e. different models can extract different information), but also that even individual ‘layers’ in a model learn differently from the same data based on how deep they are (i.e. learning and memorisation can be localised to specific parts of the model).*
> >
> > …
> >
> > *Another line of work has previously explored the issue of localisation of individual learning concepts (including memorisation and generalisation) relies on the tools from the domain of mechanistic interpretability [1,2,3]
> > In [2] it was shown that using these tools it is possible to identify three distinct stages of ML training: memorisation, circuit formation (i.e. generalisation) and clean-up (i.e. replacing the memorised information with generalised patterns).
> > This work contradicts the findings of [4], arguing that memorisation occurs prior to generalisation, showing that while memorisation can often be temporarily localised, it is not yet possible to concretely determine when it takes place.*
> >
> > *We note, however, that while the aforementioned works have made significant progress in this field, there is, yet, no clear definitive consensus on how and where to attribute individual learning concepts (and, hence, memorisation) to.*
> >
> > [1] - Hernandez, Danny, et al. "Scaling laws and interpretability of learning from repeated data." arXiv preprint arXiv:2205.10487 (2022).
> >
> > [2] - Nanda, Neel, et al. "Progress measures for grokking via mechanistic interpretability." arXiv preprint arXiv:2301.05217 (2023).
> >
> > [3] - Pearce, Adam, et al. "Do machine learning models memorize or generalize." People+ AI Research (2023).
> >
> > [4] - Arpit, Devansh, et al. "A closer look at memorization in deep networks." International conference on machine learning. PMLR, 2017.
> >
> > >7.2 - I would draw a stronger distinction between adversarial examples and data poisoning (which you don't actually name but seems to be some of what you're describing)
> >
> > We thank the reviewer for this suggestion. We agree with the reviewer on this, but we also want to highlight to the reader that data poisoning is a malicious application of adversarial samples and that these terms are very closely related.
> >
> > We have now amended Section 7.2 which reads as follows:
> >
> > *They are, however, still valid (albeit atypical) data points and can be used to train a well-generalised model. The same concepts are exploited in works on data poisoning, where a malicious actor artificially generates data points from the low-density region of the distribution, which are crafted to degrade the performance of the model.
> > These data points produced during this process are known as adversarial samples [1] and these were originally designed to showcase how tiny invisible input perturbations can be used to cause an image classification model to mispredict, but were later adapted to other settings and modalities [2].*
> >
> > [1] - Goodfellow, Ian J., Jonathon Shlens, and Christian Szegedy. "Explaining and harnessing adversarial examples." arXiv preprint arXiv:1412.6572 (2014).
> >
> > [2] - Tian, Zhiyi, et al. "A comprehensive survey on poisoning attacks and countermeasures in machine learning." ACM Computing Surveys 55.8 (2022): 1-35.

---

> > > ### Author Response · Authors · 2024-07-04
> > > **Response to reviewer PFHo (7)**
> > >
> > > >"since pixels (on their own) are independent" - I don't understand this claim, I wouldn't say that pixels are independent in a natural image
> > >
> > > We thank the reviewer for this comment. We agree that our choice of wording here was not entirely accurate. We wanted to convey that the individual pixels on their own are not necessarily meaningful (i.e. semantics are often unaffected when a single pixel is altered), compared to groups of pixels (i.e. the features).
> > >
> > > We have now corrected this and the Section reads as follows:
> > >
> > > *However, one may argue that since pixels (on their own) are not necessarily meaningful (i.e. change in a single pixel is unlikely to affect the semantic meaning of the image), but the features within natural images are, without further contextualisation most pixel-based metrics are meaningless.*

---

### Author Response · Authors · 2024-07-04
**Response to all reviewers (1)**

We would like to thank the reviewers for their valuable feedback! These suggestions and comments have really helped to improve the manuscript.

We begin by addressing the main point of concern described by all reviewers: our argument behind the choice of preferred definition for memorisation. Then we provide responses to the individual comments, highlighting reviewer’s comments in blockquotes and the changes to the manuscript in italics.

Choice of definition:

The term memorisation has been used very loosely in previous works to indicate that models can fit data with noisy labels (Zhang et al. [1]) or that they can reproduce their inputs verbatim (Carlini et al. [2]). Also, it has been used synonymously with overfitting (Mehta et al., [3] Leino et al. [4] etc.)
However, in the work by Feldman [5] it is argued that this definition is more nuanced:

Contrary to overfitting, which denotes an inability of a model to generalise, memorisation is beneficial (required even) for generalisation in the case where the data generating distribution is long-tailed, i.e. atypical examples make up a lot of the data (which can often be the case in the so-called ‘critical applications of AI’ recently recognised under the EU AI act framework)
This mirrors the "noisy label" idea in the sense that the association between inputs and labels can be tenuous for atypical examples, but also goes beyond it because even correctly labelled atypical examples can exhibit features which are not typical of their class

We note that it is not the specific definition we concentrate on, but rather the intuition behind it. Feldman's work also points to a broader, conceptual understanding that is (informally): memorisation is a phenomenon whereby the specific information from individual samples (with an emphasis on individual) rather than the general information that can be learned from any number of generic samples is influential/meaningful to a model's training.

Corroborating evidence for memorisation and overfitting being distinct is given by Carlini et al. [2] who claim that memorisation occurs before overfitting (or overtraining as it is referred to in [2]). A model’s "likelihood" at a specific sample can spike very early, if the example has enough influence, i.e. the ability to ‘bend’ the model to its features. We mean the word "bend" quite literally here, as there is evidence (Garg et al. [6]) that high influence samples cause the curvature of the loss landscape to increase and/or cause significant distortion in the decision boundary. Influence functions essentially measure the first of these phenomena.

Feldman makes their formal argument about memorisation by measuring the distances between label distributions. This implies supervised classification as the chosen ML setting. A more general approach was proposed as the follow-up work by Zhang et al. [7] and is termed ‘counterfactual memorisation’, which just uses a different test function and takes an expectation, and generalises Feldman's [5,8] definition to semi-supervised settings, regression, etc. Regardless, the notion of an example strongly influencing the model's behaviour remains.

We believe that these are good definitions because they are formally justified in long-tailed data distribution settings, which most tasks are [5], and because (generalised) self-influence is flexible enough to accommodate a variety of learning schemes. There is also additional empirical evidence from Feldman [5,8] Zhang et al.[1,7] that the theoretical predictions actually transfer to real datasets.

The connections to stability and to DP are pretty immediate from the definition (see Feldman [5] on the relationship between memorisation and leave-one-out stability), and from Brown et al. (2021) [9]: Informally, stable algorithms memorise less, and DP algorithms are "very stable".

The definition we select is thus very close to "memorisation is the phenomenon that DP limits". Why do we argue that this makes it a good definition? Because generating copyrighted content and harming people (privacy, fairness etc.) are two of the biggest security/safety risks of current AI systems, and it is important to measure and suppress them in a strong/formal way. Extractability, exposure and other related terms are mostly empirical/heuristic, and methods which do not measure memorisation directly, but using some proxy instead, can yield misleading conclusions, or only work for specific models or datasets.
In addition to that, we note (similar to [2,5,7,8]) that high memorisation of some instances does not make it a problem by itself, as certain data samples should, in fact, be memorised. These include examples such as common facts, quotes from books, algorithms, etc. - which further shows that memorisation can have a positive impact on ML models.

---

> ### Author Response · Authors · 2024-07-04
> **Response to all reviewers (2)**
>
> This intuitively explains the choice of terminology (humans have to memorise "special" facts which don't fit into "general knowledge"), the connection to individual influence and to notions like DP and leave-one-out generalisation (these restrict/quantify the reliance of models on individual samples). Methods which do not directly measure memorisation (but the things leading to it) all seem to leverage this fact: the samples which fall under this behaviour are distinct and individual, (or "special"). This makes Feldman a very attractive definition to describe the phenomenon of memorisation.
>
>
> [1] - Zhang, Chiyuan, et al. "Understanding deep learning (still) requires rethinking generalization." Communications of the ACM 64.3 (2021): 107-115.
>
> [2] - Carlini, Nicholas, et al. "The secret sharer: Evaluating and testing unintended memorization in neural networks." 28th USENIX security symposium (USENIX security 19). 2019.
>
> [3] - Mehta, Harsh, Ashok Cutkosky, and Behnam Neyshabur. "Extreme memorization via scale of initialization." arXiv preprint arXiv:2008.13363 (2020).
>
> [4] - Leino, Klas, and Matt Fredrikson. "Stolen memories: Leveraging model memorization for calibrated {White-Box} membership inference." 29th USENIX security symposium (USENIX Security 20). 2020.
>
> [5] - Feldman, Vitaly. "Does learning require memorization? a short tale about a long tail." Proceedings of the 52nd Annual ACM SIGACT Symposium on Theory of Computing. 2020.
>
> [6] - Garg, Isha, and Kaushik Roy. "Memorization through the lens of curvature of loss function around samples." arXiv preprint arXiv:2307.05831 (2023).
>
> [7] - Zhang, Chiyuan, et al. "Counterfactual memorization in neural language models." Advances in Neural Information Processing Systems 36 (2023): 39321-39362.
>
> [8] - Feldman, Vitaly, and Chiyuan Zhang. "What neural networks memorize and why: Discovering the long tail via influence estimation." Advances in Neural Information Processing Systems 33 (2020): 2881-2891.
>
> [9] - Brown, Gavin, et al. "When is memorization of irrelevant training data necessary for high-accuracy learning?." Proceedings of the 53rd annual ACM SIGACT symposium on theory of computing. 2021.

---

### Decision · Action_Editor_kA1a · 2024-08-05

**Recommendation:** Accept as is

**Comment:**

This is a solid survey of memorization techniques. The initial version of the paper had some overly optimistic claims, but the current version after reviewer feedback has toned it down. I support acceptance.

**Audience:**

The audience would be the section of the AI community that is interested in privacy and memorization measurement.

**Claims And Evidence:**

This paper is a survey of memorization techniques, and the claims appear to be well supported.